# Dreaming to Assist: Learning to Align with Human Objectives for Shared Control in High-Speed Racing

**Jonathan DeCastro⋆, Andrew Silva⋆, Deepak Gopinath, Emily Sumner, Thomas M. Balch, Laporsha Dees, Guy Rosman**

Toyota Research Institute, Cambridge, MA, USA, `first.last@tri.global`
⋆ Contributed equally

**Abstract:**
Tight coordination is required for effective human-robot teams in domains involving fast dynamics and tactical decisions, such as multi-car racing. In such settings, robot teammates must react to cues of a human teammate's tactical objective to assist in a way that is consistent with the objective (e.g., navigating left or right around an obstacle). To address this challenge, we present DREAM2ASSIST, a framework that combines a rich world model able to infer human objectives and value functions, and an assistive agent that provides appropriate expert assistance to a given human teammate. Our approach builds on a recurrent state space model to explicitly infer human intents, enabling the assistive agent to select actions that align with the human and enabling a fluid teaming interaction. We demonstrate our approach in a high-speed racing domain with a population of synthetic human drivers pursuing mutually exclusive objectives, such as "stay-behind" and "overtake". We show that the combined human-robot team, when blending its actions with those of the human, outperforms the synthetic humans alone as well as several baseline assistance strategies, and that intent-conditioning enables adherence to human preferences during task execution, leading to improved performance while satisfying the human's objective.

**Keywords:** Recurrent State-Space Models, Human-Robot Interactions, Shared-Control

## 1 Introduction

In high-stakes situations where members of a team must coordinate their physical actions in the world for the team to succeed, early coordination on tactical objectives is crucial. In a rapidly-evolving task, such as in high-speed competitive sports, agents must find a way to attain such coordination without explicit communication. A robotic assistive agent equipped with an ability to reason using theory of mind [1] has been shown to be critical to successful collaboration without the burden and latency of explicit communication. In such settings, agents must maintain a rich-enough model of the world to cover a common set of concepts that each agent needs to plan. This includes dynamics of the physical world, the objectives of the team, and the current intent of the other members of the team. Such considerations are prevalent in sports [2, 3], manufacturing [4], healthcare [5], and traffic modeling settings [6], among others.

High-speed performance driving presents a domain where accurate and expressive models are required to have effective human-robot teams. The dynamics in racing evolve quickly, preventing team members from communicating their goals or objectives before taking an action [7]. Because of this constraint, existing approaches in shared control or advanced driver assistance often split authority on predefined boundaries (e.g. steering vs. throttle and brake) [8, 9]. The driving domain also requires us to tackle the multimodal nature of the human decision-making problem (e.g. rules or maneuvers),

8th Conference on Robot Learning (CoRL 2024), Munich, Germany.

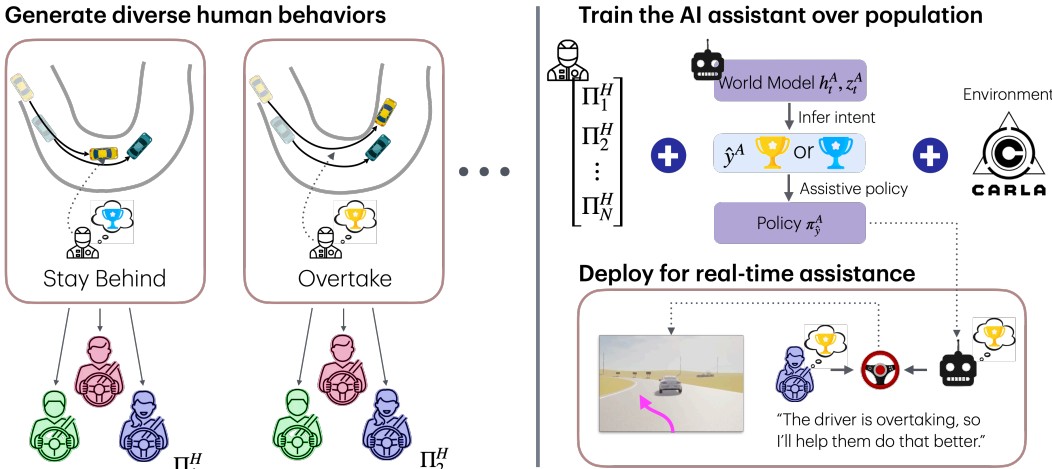

**Figure 1:** We mimic human preferences on discrete decisions via population clusters during world model formation. Our assistive models then learn to decisively help on the overall discrete-continuous control task while taking into account multiple possible human preferences.

where discrete decisions are required to be made given knowledge of the situation [10, 11, 12, 13]. We hypothesize that a theory-of-mind-inspired model, coupled with recent advances in flexible planning and reinforcement learning, can enable new ways for machines to help humans drive more efficiently and effectively. We view our work as complementary to existing works in game-solving for racing [14] and planning in racing [15].

In this paper, we present a modeling paradigm to explicitly infer and support a human teammate's intent, enabling our agent to learn how to interact with diverse human strategies in a high-speed continuous control task. We consider the problem of a robotic assistive agent in a race car that helps a human driver perform tactical overtaking maneuvers more safely and more optimally in a high-speed competitive domain where explicit communication is difficult. While the driver remains in control of the vehicle at all times, the assistive agent can augment the driver's lateral (steering) and longitudinal (throttle and brake) actions. The assistant must infer the driver's intention (e.g., whether or not the driver is going to attempt an overtake) in order to provide optimal control augmentations and modifications that help the driver execute the inferred intention.

We address planning under the unobservable dynamics involved with human decisions acting in the world via a novel model-based reinforcement learning (MBRL) paradigm that jointly learns the dynamics of the physical world and the human's intentions, enabling fluid shared control with continuous assistance during a race. We contribute the learning of a recurrent state-space model within an MBRL setup in light of cognitive and neuroscience findings indicating human decision-making may be modeled as such [16]. We infuse a world model with a richer world-representation by using fictitious humans with multiple, mutually exclusive, objectives with the aim of encoding diverse human intents into the model's world representation. Specifically, we contribute:

1. A novel approach to MBRL for shared control in a tactical setting, leveraging driver intent modeling using fictitious co-play, conditioned on a fixed set of human objectives.

2. A means for expert human value alignment, which conditions an assistive agent's rewards based on inferred intent to reason jointly over the physical world and the human's behavior.

3. An evaluation of our approach on a shared control racing domain, demonstrating our model's utility over a diverse set of imperfect fictitious humans with different personal objectives.

## 1.1 Related Works

Significant work has taken place in sharing control [8] and human-robot interaction [17, 18], exploring topics from ergonomics and physiology to language-based strategic teaming. Prior research has

presented learning-based approaches to human-robot interactions that target joint-representation learning [19, 20, 21, 22, 23, 24, 25, 26]. These works focus on a range of topics, including data-efficient representation learning [27], intent or plan modeling [28, 29, 30, 31], hypothesis-space specification and explanation [32, 24, 33], or other cognitive and social motivations [34, 35, 36]. We go beyond prior work by inferring the human's discrete intent and modifying our robotic assistant's objective to encourage support of the inferred intent.

In the context of driving, shared control has been explored for planning approaches [37, 38, 39, 40, 41, 42, 43, 44, 45] and learning-based approaches [46, 47, 48, 49, 50]. Prior work has also considered incorporating game-theory into shared control [50, 51], as well as explicit human-centric design considerations [52, 53]. We refer readers to [54, 55] for comprehensive reviews. Beyond shared control, significant literature explored intent prediction in driving, see [56, 57], and references therein.

Recent work has proposed a shared-control model using model-predictive control that considers predicted trajectory information [39], therefore implicitly capturing driver intent. Additional recent work has augmented a model-predictive controller with the ability to explicitly infer driver intent [9], enabling the controller to share steering and control actions with a human. In our approach, we capture discrete driver intent in a way that is conducive to semantically-meaningful, multi-modal continuous behaviors (e.g., going left *or* right around an obstacle). Further, by framing our task as a multi-agent reinforcement-learning problem, our proposed solution extends to multiple agents much more naturally than prior work. Finally, we frame our decision making approach within a recurrent state-space model [58] which is extended to infer the objective of the human, building on recent work that has explored hierarchical or hybrid state abstractions [59, 60].

## 2   Background and Problem Statement

We target the problem of shared control in the highly-dynamic setting of high-performance racing against other racing opponents. We aim to build an assistive agent that is capable of reasoning over well-defined *task objectives*, as well as more general, harder-to-define, and harder-to-observe *human objectives*. The assistive agent is given a map of the track, states of the ego vehicle and opponent vehicle, the ego driver's steering and throttle controls. The agent's task is to assist the ego driver through modifications to the steering and throttle of the ego car, as depicted in Fig. 1 (lower right).

To achieve optimal performance, the agent must provide continuous control adjustments as the driver progresses along the track, helping the driver to stay on course, maintain proper speeds, and avoid collisions. Further, the agent must accurately infer the intentions of the driver (such as "stay to the opponent's left") and provide control augmentations that help the driver to accomplish their immediate task objective more optimally. Note that this problem is different from a conventional autonomous driving problem, as the agent's actions are conditioned explicitly on the human driver's control input, and the control signal that actuates the car is a linear blend of the agent and the driver's control signals. This problem also differs from conventional human-robot teaming, in that the agent must infer a human's intent (i.e., the human objective) early and as consistently as possible, as the efficacy on its assistance (in terms of safety and performance) depends strongly to how early, accurately, and reliably the human objective is captured.

## 3   Approach

Our approach learns a common latent representation in the structure of a recurrent state space model (RSSM) [61, 62] to build a hybrid discrete / continuous state of the human partner's behavior, their rewards, and intent for achieving certain goals. RSSMs have been shown to be suitable for many domains (e.g. locomotion, Atari, Minecraft), and we are the first to apply this to modeling human behavior for assistance. To accommodate both a *human* and *assistive* agent operating in a collaborative setting, we train the assistive agent alongside human agents, with both agents sharing actions taken in an environment. Each agent receives a reward for each action taken, and each are trained to (1) build an accurate RSSM and (2) learn how to act in order to maximize its own expected returns [63]. One challenge in collaboration is that part of the assistive agent's world model includes

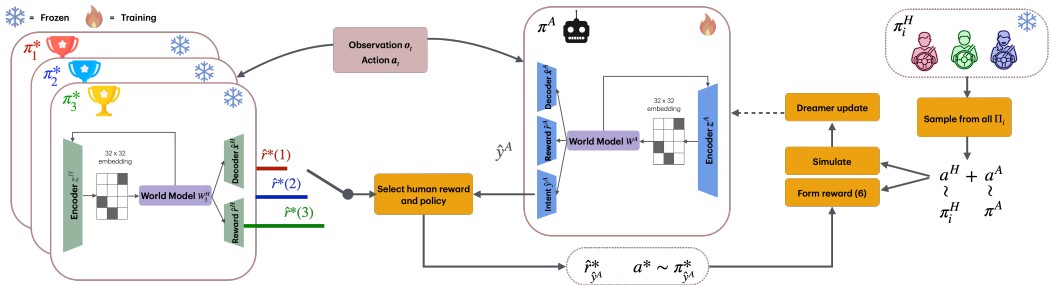

**Figure 2:** Overview of value alignment for the assistive agent. We start with a set of frozen human policies whose values are annotated with predetermined outcomes. We blend human actions linearly with the actions from an unfrozen assistive agent. For both human and assistive agents, a multi-head RSSM architecture is then used to predict the observations, reward, and human intent (assistive only), trained to maximize their log-likelihoods against samples taken from the environment. The intent head is trained to match frozen human intents, which are fixed a priori. The assistive agent's rewards are shaped based on the *optimal policy* $\pi^*_{\hat{y}^A}$ and optimal predicted reward $r^*_{\hat{y}^A}$ for inferred intent $\hat{y}^A$.

aspects of the future human's plan such as preferences, desires, and goals of the human (these are often collectively referred to as objectives [32] or intents [30] in the literature, we will use these terms interchangeably in the paper), which are often unknown and only weakly observed by their behavior [32]. We extend the RSSM formulation to allow it to be supervised on a diverse set of humans, whose objectives (intents) are known and labeled, to force the representation to be jointly aware of the behavior and intent across a variety of human types and physical environments. We then devise a scheme that feeds in inference of the human's objective into the assistant's reward function.

### 3.1 Building a World Model over the Human and Physical Environment

In order to allow the system to maintain an ability to reason over the preferences of the human independent of that of the joint human-robot system, our approach considers both the human driver and the assistive agent planner as separate models. The training process is outlined in Fig. 2.

In the cooperative setting, the model of the environment follows a structure in which there are certain task-specific rewards which are available to both human and assistive agents, with the key distinctions being human-objective (specific to the human), and intervention penalties (specific to the assistant).

**Task-Specific Rewards** We assume standard task-specific rewards for high-performance driving from prior work [64, 15], including an out-of-bounds penalty, passing reward, and collision penalty.

$$r_t^{task} = r_t^{collision} + r_t^{bounds} + r_t^{finish} \tag{1}$$

where $r_t^{collision}$ is a negative reward for collisions, $r_t^{bounds}$ is a negative reward for driving too far off the track, and $r_t^{finish}$ is a positive reward for reaching the finish line.

**Intent-Aware World Model** The objective of the world model is to provide a representation that the agent can use to interact with the driver and the world, and we posit that this representation can support the agent to reason jointly about both the task and human objectives. We build off of the recurrent state-space model (RSSM) of DreamerV2/V3 [65, 61], according to the architecture in Figure 2. For agent $\kappa \in \{H, A\}$ (respectively, the **H**uman and **A**ssistive agent), the RSSM model, parameterized by $\phi$ and denoted $W_\phi^\kappa$, includes:

$$
\begin{array}{lll}
\text{Encoder for discrete representation } \boldsymbol{z}_t^\kappa: & \boldsymbol{z}_t^\kappa \sim q_\phi^\kappa(\boldsymbol{z}_t^\kappa | \boldsymbol{h}_t^\kappa, \boldsymbol{x}_t) & \\
\text{Sequence model for recurrent state } \boldsymbol{h}_t^\kappa: & \boldsymbol{h}_t^\kappa = f_\phi^\kappa(\boldsymbol{h}_{t-1}^\kappa, \boldsymbol{z}_{t-1}^\kappa, \boldsymbol{a}_{t-1}^\kappa) & \\
\text{Dynamics predictor:} & \hat{\boldsymbol{z}}_t^\kappa \sim p_\phi^\kappa(\hat{\boldsymbol{z}}_t^\kappa | \boldsymbol{h}_t^\kappa) &
\end{array} \tag{2}
$$

where $\boldsymbol{x}_t$ denotes the input observation. The output heads are similar to the DreamerV2/V3 architecture, with the addition of an intent predictor for the assistant agent (in red), and are all bottlenecked on the hidden states $\boldsymbol{s}_t^\kappa = \{\boldsymbol{h}_t^\kappa, \boldsymbol{z}_t^\kappa\}$,

$$
\begin{array}{llll}
\text{Decoder:} & \hat{\boldsymbol{x}}_t^\kappa \sim p_\phi^\kappa(\hat{\boldsymbol{x}}_t^\kappa | \boldsymbol{s}_t^\kappa) & \text{Continue predictor:} & \hat{c}_t^\kappa \sim p_\phi^\kappa(\hat{c}_t^\kappa | \boldsymbol{s}_t^\kappa) \\
\text{Reward:} & \hat{r}_t^\kappa \sim p_\phi^\kappa(\hat{r}_t^\kappa | \boldsymbol{s}_t^\kappa) & \text{Intent predictor:} & \hat{y}_t^A \sim p_\phi^A(\hat{y}_t^A | \boldsymbol{s}_t^A)
\end{array} \tag{3}
$$

To train the intent predictor, we assume at least partial access to the intent labels for a given episode, which takes the form of an integer-valued target function. We normalize the output by predicting the symlog $(\text{sign}(y)\ln(|y|+1))$ of the output, and use a discrete distribution to predict $y$ using the two-hot encoding of [61]. The parameters $\phi$ of the world model $W_\phi^\kappa$ are trained to minimize the loss

$$\mathcal{L}^\kappa(\phi) = \mathbb{E}_{q_\phi^\kappa}\left[\beta_{pred}\mathcal{L}_{pred}^\kappa + \beta_{KL}\mathcal{L}_{KL}^\kappa\right]$$

The prediction loss $\mathcal{L}_{pred}^\kappa$ minimizes the likelihood under the predictor distributions in (3), $\mathcal{L}_{KL}^\kappa$ minimizes the KL divergence between the prior $p_\phi^\kappa$ and the approximate posterior $q_\phi^\kappa$, and $\beta_{pred}$ and $\beta_{KL}$ are scalar weighting values. Given an intent label $y_t$, the portion of loss $\mathcal{L}_{pred}^A$ for the intent predictor is taken as the negative log-likelihood of the label under $p_\phi^A$.

The world model training is alternated with training for the behavior model governing actions $\boldsymbol{a}_t^\kappa$, with the behavior model learned via an actor-critic policy training over the estimates $\boldsymbol{s}_t, \hat{r}_t$,

$$\text{Actor:} \qquad \boldsymbol{a}_t^\kappa \sim \pi_\theta^\kappa(\boldsymbol{a}_t^\kappa \mid \boldsymbol{s}_t^\kappa) \qquad \text{Critic:} \qquad v_\psi^\kappa(\boldsymbol{s}_t) \approx \mathbb{E}_{p_\phi^\kappa, \pi_\theta^\kappa}[\hat{r}_t^\kappa] \qquad (4)$$

The critic is trained to minimize the temporal-difference loss on the value function $v_\psi^\kappa$. The actor attempts to maximize the critic-predicted value. For further details, we refer the reader to [65].

## 3.2 Alignment with the Human

To learn alignment with human drivers, we expose the assistive agent to different humans at training time. Training with human partners in a fictitious co-play setting [27] allows the assistant to become robust to different possible human behaviors, but does not teach the assistant to distinguish between the discrete modes of human behavior, which is necessary for assistance in our driving task.

Similar to recent work describing human preferences for a task using reward shaping [66, 67], we generate a population of humans, but different from these works, we group labeled sub-populations according to certain inherent objectives or preferences. We train several sub-populations using a separate human reward $r_t^H = r_t^{task} + r_t^y$, which is composed of the base task rewards in (1), with the addition of several distinct *human objectives*, spanning different multimodal behaviors of the human. These are then formed into a collection of humans $\{\pi_1^H, \pi_2^H, \ldots, \pi_N^H\} \in \{\Pi_1^H, \Pi_2^H, \ldots, \Pi_N^H\}$ for $N$ different objectives. Examples of different settings for $r_t^y$ are given in Appendix C.

If trained to optimality, such rewards induce optimal behaviors for each human objective $\{\pi_1^*, \pi_2^*, \ldots, \pi_N^*\}$, which we can consider as the collection of *expert policies* of each enumerated objective. We extend this notion to that of *intent* at runtime. At any given moment, the human may adopt an intent $y$ to abide by policy $\pi_y^H$. To simplify training, we consider drivers and rollouts with fixed intents, $y$, though our problem setup generalizes to variable-intent scenarios (e.g., a human switching from "following" to "passing" during a race).

Humans are error-prone and pursue their objectives irrationally [68]. To capture this, we partially train human partners to pair with the assistive agent in a process known as fictitious co-play (FCP) [27], yielding a population of humans $\{\pi_{y,k}^H\}_k \cup \{\pi_y^*\}$ for each intent sub-population $\Pi_y^H$, where $k$ represents the policy checkpoint for some amount of completed training. Using the FCP framework, each checkpoint $k \in [1, K]$ is an agent trained to $k\%$ completion and the final checkpoint $k = K$ is an optimal policy. Each (sub)optimal human is endowed with a (sub)optimal RSSM world model. While we apply FCP in this work, we note that our approach is agnostic to how human behaviors are generated and future work may consider other approaches to synthesizing human policies [69, 70].

## 3.3 Assistive Agent Objectives

For continuous actions $\boldsymbol{a}_t^H, \boldsymbol{a}_t^A \in \mathbb{R}^m$ for the human and assistant, respectively, we encourage the assistive agent to minimize its action intervention by adopting a reward of the form

$$r_t^{interv} = -\|\boldsymbol{a}_t^A\|_2$$

which penalizes the magnitude of the intervention according to an $L_2$-norm. Aggregated over time, the intervention cost forms a mixed-norm $L_1 - L_2$ [71], encouraging time sparsity of interventions.

In addition to the intervention norm penalty, the agent is rewarded for task completion and penalized for driving out of bounds or for collisions. These sparse negative rewards serve to encourage the assistant to act as a "racing guardian", keeping the driver on the track and avoiding collisions while making progress towards the finish line. Note that these rewards align with the *task* objective (Sec 3.1), but do not favor any particular *human* objective or intent.

Finally, the assistive agent receives reward for aligning with the inferred objective of the *human driver*. First, the agent receives a reward equal to the inferred reward under the optimal human's world model. In other words, at every time step $t$, the agent predicts the human's intent, $\hat{y}_t^A$ and then queries the world model corresponding to the optimal human policy $W_{\phi(\hat{y}_t^A)}^*$ for its reward prediction $\hat{r}^*(\hat{y}_t^A)$, for the current state-action pair. Second, the agent receives reward for maximizing the likelihood of the human-robot's combined action under the optimal human model for the predicted human intent:

$$\pi_{\hat{y}_t^A}^*(\boldsymbol{a}_t^A + \boldsymbol{a}_t^H | \boldsymbol{x}_t) := \pi^*(\boldsymbol{a}_t^A + \boldsymbol{a}_t^H | \boldsymbol{x}_t, \hat{y}_t^A) \tag{5}$$

Note that $\boldsymbol{a}_t^H$ is the current sub-optimal human action. Intuitively, these rewards encourage the assistant to: (1) select actions for which the optimal human assigns high value and (2) select actions that bring the sub-optimal human closer to the optimal human. Under this setup, although we infer $\hat{y}_t^A$ from the joint human-robot trajectory, we consider it to be more aligned with the *human's* intent, due to: (1) the presence of the intervention sparsity term $r_t^{interv}$, and (2) the fact that the inferred expert alignment term tilts the joint behavior toward the inferred expert human's behavior.

The complete reward for the assistant is a combination of rewards that promote successful task execution, minimize the magnitude of intervention, and account for human preferences:

$$r_t^A = \underbrace{r^{task} + r_t^{interv}}_{\text{task performance and intervention sparsity}} + \underbrace{\alpha_r \hat{r}_t^*(\hat{y}_t^A) + \alpha_a \|\boldsymbol{a}_t^H + \boldsymbol{a}_t^A - \boldsymbol{a}_t^*(\hat{y}_t^A)\|_2}_{\text{expert alignment}} \tag{6}$$

Here, scalars $\alpha_r$ and $\alpha_a$ weight the contributions of the human terms. We capture both the inferred optimal reward value, $\hat{r}^*(\hat{y}_t^A)$, as well as the similarity between the combined human-robot actions and an action from the inferred optimal policy $\boldsymbol{a}^*(\hat{y}_t^A) \sim \pi_{\hat{y}_t^A}^*(\boldsymbol{x}_t)$. Note that, though we empirically compare different values of $\alpha_r$ and $\alpha_a$ in Appendix E, in the next section, we only examine results for the reward-only case ($\alpha_r = 1, \alpha_a = 0$).

## 4 Experiments and Results

We examine the performance of our approach, dubbed DREAM2ASSIST, on different racing tasks with a fictitious human driver. We examine two racing settings, each derived from portions of a two-mile race track, and implemented in the CARLA Simulator [72] using a rear-wheel race vehicle physics model. For each task, opponents are randomly instantiated as replays of real human trajectories from the track, meaning that they do not react to the ego vehicle. In each setting, we run two sets of experiments—one with *pass vs. stay* fictitious human partners (i.e., the human is trying to overtake or stay-behind their opponent), and one with *left vs. right* partners (i.e., the human wants to stay on the left or right side of their opponent). We also further examine out-of-distribution intent inference (with intent changing over time), and further ablations in Appendix F and G.

In each experiment, we train a population of humans following each objective (e.g., "left" or "right"), and then train a DREAM2ASSIST agent over the combined population. We then evaluate the degree to which the assistive agent can improve fictitious human performance on the track, where performance is measured by total progress, average speed, and no collisions. To measure the contributions of each assistive agent for the fictitious human population, we sample checkpoints at every 20% performance increment for agents up to at least 75% of maximum, or from the bottom five performers if none are under this threshold. We report the mean change in performance (track **progress** and **collisions**) when an assistive agent is deployed, as well as the **return** under each human objective in Table 1. Means and standard deviations are computed over all five sub-optimal fictitious drivers.

The two settings we consider include **Straightaway Driving** and **Hairpin Driving**. The straightaway is a flat 370-meter portion of a track with a concrete barrier on the right-hand side. Because the

**Table 1:** Results on straightaway and hairpin experiments. Blue indicates improvement, **bold** is best.

| | Pass (top) / Stay (bottom) | | | | | | Left (top) / Right (bottom) | | | | | |
|---|---|---|---|---|---|---|---|---|---|---|---|---|
| | Progress ↑ | Hairpin Return ↑ | Collisions ↓ | Progress ↑ | Straightaway Return ↑ | Collisions ↓ | Progress ↑ | Hairpin Return ↑ | Collisions ↓ | Progress ↑ | Straightaway Return ↑ | Collisions ↓ |
| Dreamer | -11.3±13.6 | **0.5±2.9** | -0.1±0.2 | -0.7±4.6 | **-0.1±0.9** | 0.0±0.1 | 21.8±28.7 | -2.0±2.2 | 0.1±0.1 | **10.8±7.3** | -0.6±0.3 | 0.0±0.1 |
| | -21.1±68.6 | 0.3±0.4 | **0.0±0.0** | -6.1±17.5 | **0.0±0.1** | 0.1±0.2 | 10.0±17.9 | -1.3±1.8 | 0.0±0.1 | -1.1±6.2 | **0.2±1.2** | 0.0±0.1 |
| Dreamer-AIL | -28.9±46.2 | -5.7±7.3 | **-0.3±0.2** | -116.6±58.7 | -9.7±4.2 | **-0.4±0.3** | -144.0±89.9 | -7.1±6.6 | **-0.4±0.2** | -134.4±13.0 | -2.5±0.9 | **-0.5±0.1** |
| | -216.6±145.6 | -1.7±0.4 | 0.1±0.2 | -52.5±45.9 | -1.6±0.2 | **-0.1±0.1** | -119.1±79.9 | -4.5±17.3 | **-0.4±0.3** | -126.3±53.2 | -13.4±7.0 | **-0.5±0.2** |
| Dream2Assist | **71.8±43.9** | -1.2±3.3 | -0.2±0.2 | **6.0±9.1** | -0.4±1.4 | **-0.1±0.1** | **54.8±60.4** | **1.4±5.2** | 0.0±0.1 | 5.0±5.0 | **0.1±0.5** | 0.0±0.1 |
| | **60.5±49.7** | **0.8±0.4** | 0.1±0.1 | **57.8±36.4** | -0.1±0.2 | 0.1±0.1 | **27.2±23.1** | **2.2±2.3** | -0.2±0.2 | **-1.1±31.91** | -2.8±2.4 | 0.2±0.2 |

section of road is straight and flat, drivers need only avoid colliding with the concrete wall or their opponent while racing to the finish. The hairpin section, however, is a 960-meter portion of a track involving several sharp turns and hills. Drivers must carefully manage their speed to avoid spin-outs or collisions, and passing is much more challenging in this section of track.

**Human Decision Characteristics** Based on observed behaviors from a human study (see Appendix B), we propose two different sets of human characteristics (ground truth intent): **pass vs. stay** and **left vs. right**. We train fictitious human agents to satisfy each of these objectives. For the *pass vs. stay* behaviors, the fictitious human agents are trained to either overtake or stay-behind the opponent vehicle. For the *left vs. right* behaviors, the fictitious humans are trained with a preference to stay on either the left or right side of the opponent for as much of the race as possible. Attempting to provide the wrong type of assistance with such distinct behaviors (e.g., offering "right" assistance to a "left" human) will result in fighting between the assistant and human, and will likely lead to collisions or spin-outs. We provide further details on the rewards for each agent in Appendix C.

**Action and Observation Spaces** The environment itself consists of the rewards in Sec. 3.1 and an observation and action space for each agent. The human and assistive agent's actions both adopt steering and acceleration values in the range of $[-1, 1]$. The assistant and human actions are summed together and clipped on $[-1, 1]$ before being passed to the vehicle. The observation space contains the ego vehicle state (position, velocity, tire slip angles, yaw rate, heading), current distance traveled, and an array of forward-looking track edge points. Finally, the human and assistive agents are each able to observe the other's actions.

**Baselines** The claim of our work is that an assistive agent will better support a human partner if the assistant can infer the human's objective and help to satisfy that objective. To test this claim, we compare **DREAM2ASSIST** against a baseline version of **DREAMER**, which makes no intent inference and is not rewarded according to a human objective. We also compare to a non-RSSM baseline, GAIL [73], obtained with the BeTAIL [74] framework. BeTAIL uses a behavior transformer as the human policy, coupled with an assistive agent trained via adversarial imitation learning to correct for distribution shift. Our **DREAMER-AIL** baseline challenges whether our RSSM approach is necessary at all, or whether a purely data-driven approach using behavior cloning and inverse reinforcement learning could satisfy the multimodal behavioral assistance task that we consider. For this baseline, a fictitious human using a DREAMER policy is paired with the AIL assistive agent from BeTAIL.

## 4.1 Straightaway Results

We observe a consistent trend for all straightaway results – DREAMER offers very low-magnitude intervention, leading to higher performance with expert drivers but poor performance with novice drivers (as the assistant is not helping). We show track progress and human-objective return for assistants deployed to the *pass vs. stay* problem in Fig. E.1 (bottom). We report additional results in Appendix E. Conversely, DREAM2ASSIST offers higher-magnitude intervention, leading to higher performance with novice drivers, but lower performance gains with expert drivers (as the assistant is not needed). As shown in Table 1, DREAM2ASSIST generally offers the highest performance over other methods. DREAMER-AIL suffers from mode collapse, trying to turn all drivers into either a "stay" driver or getting caught between "left" and "right" and therefore not moving. This behavior means that the DREAMER-AIL agent consistently underperforms a fictitious human with no assistance at all, as the AIL assistance keeps the driver far behind the opponent.

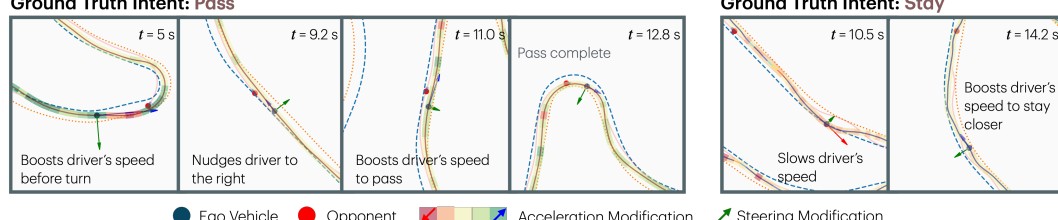

**Figure 3:** Examples of the DREAM2ASSIST agent's actions when paired with a human intending to pass and a human intending to stay. DREAM2ASSIST recognizes the driver's intent, making lateral corrections for a safer overtake (left) or throttle adjustments to stay behind the opponent while still progressing towards the finish (right), thereby helping to satisfy *task* and *human* objectives.

## 4.2 Hairpin Results

The hairpin domain is more challenging and yields fictitious humans that are not always consistently able to solve the task, thereby leaving greater scope for assistance from our trained agents. In our *pass vs. stay* experiment, DREAM2ASSIST significantly improves the drivers' abilities to solve the task while still satisfying the human objectives and not leading to an increase in collisions. Similarly, in the *left vs. right* experiment, DREAM2ASSIST leads to significant increases in track progress, reduction in collisions, and improvements in human-objective alignment. Baseline approaches fail to disentangle the "left" and "right" modes of driving. The DREAMER baseline instead opts to push the driver to stay behind the opponent vehicle in an effort to reduce collisions, thereby making it farther down the track but failing to overtake. The DREAMER-AIL baseline drives aggressively off the track, leading to a significant drop in track progress, collisions, and human-objective alignment. An illustrative example of DREAM2ASSIST is in Fig. 3, and videos are at `https://youtu.be/PVugoxqX5Co`.

Figure E.1 (top) provides an overview of the amount of assistance provided to drivers for both track progress and human-objective return for assistants deployed to the *pass vs. stay* problem. Visualizing the assistance provided to each driver, we can more easily compare the scale of assistance provided by DREAM2ASSIST compared to baselines, showing that DREAM2ASSIST provides marked improvements relative to baseline assistive agents.

## 5 Conclusions and Limitations

We introduce an assistive paradigm, DREAM2ASSIST, that learns to interact with humans to help them perform more optimally while supporting their personal objectives for the task. We evaluate DREAM2ASSIST in a dynamic and challenging task of high-speed racing, and we show that our approach is able to disentangle and accommodate distinct human objectives more effectively than baseline methods. We show that DREAM2ASSIST results in higher human-robot team performance than baseline methods, suggesting that explicit intent-conditioning and reward-inference can provide crucial performance gains in settings with multimodal, mutually-exclusive, human objectives.

While DREAM2ASSIST represents a state-of-the-art improvement in human-robot teaming, there are limitations in our work that we hope to address in future work. First, our approach has been tested with fictitious humans, but we have not yet evaluated generalization to real human-robot teams. In future work, we intend to deploy DREAM2ASSIST in a human-subjects experiment to test how effectively our framework generalizes to real, sub-optimal human drivers. DREAM2ASSIST also relies on privileged access to the inferred reward values from the RSSM of an optimal policy; future work may consider how to estimate optimal policy rewards without such a model. Finally, future work may consider how to provide assistance via multiple modalities (e.g., providing sparse language guidance on when to overtake vs. dense control-level assistance during the overtake).

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

## A  MBRL Preliminaries

We frame the model-based reinforcement learning (MBRL) problem as a two-player (human ego driver, assistive agent ego driver) partially-observable Markov decision process (POMDP) defined by the tuple $M = \langle \mathcal{X}^\kappa, \mathcal{A}^\kappa, \mathcal{T}^\kappa, \mathcal{R}^\kappa, \gamma \rangle^{\kappa=H,A}$, where, for agent $\kappa \in \{H, A\}$ (for the human, and AI agent, respectively), $\mathcal{X}^\kappa$ denotes the imagined states of the world, $\mathcal{A}^\kappa$ denotes the agent's (continuous or discrete) actions, $\mathcal{T}^\kappa : \mathcal{X}^\kappa \times \mathcal{A}^\kappa \mapsto [0, 1]$ is the transition probability, $\mathcal{R}^\kappa : \mathcal{X}^\kappa \times \mathcal{A}^\kappa \times \mathcal{X}^\kappa \mapsto \mathbb{R}$ is a reward function, and $\gamma \in [0, 1]$ is a discount factor. We aim to train both agents such that they maximize their expected returns $R^\kappa = \mathbb{E}\left[ \sum_{t=1}^T r_t^\kappa \right]$.

Crucially, in the semi-cooperative shared control setting, each reward $r_t^\kappa$ is factored into sub-components, with both sharing the same task (driving) rewards, but where $r_t^H$ contains an additional term for a human's objective, and $r_t^A$ contains additional terms to weaken its contribution in relation to the human's and enforces alignment to the human.

## B  Human Subject Data Collection

We briefly discuss a study conducted for gathering human subject behavior data in the racing domain we use in the paper. The purpose of the study was to gather qualitative and statistical data on individuals' behavior and objectives in a racing context, and to use that to inform what criteria are important for building models of human objectives. We recruited 48 participants to drive a simulator with the hairpin and straightaway segments of the two-mile track, the same domains for the computational results in this paper. The scenarios were chosen so as to present overtake opportunities in portions of the track of varying levels of difficulty, while keeping the overall task short enough to ensure there is a rich interaction between the ego and opponent. Participants completed a series of warm-up trials in each domain, with three trials devoted to the straightaway segment and eight trials in the hairpin segment, each featuring different opponents of varying difficulty (fixed trajectories) to race against. Again, these were the same trajectories used in our domains.

At the conclusion of each trial, participants answered the question: "Did you attempt to pass the other vehicle?" on an iPad. We also gathered, from trajectory data, whether or not the participant actually completed an overtake without collisions or spin-outs. These results are reported in Table B.1. We conclude that even in a simulated setting, there were a lower number of actual overtakes that occurred than were attempted. This suggests that there is room to assist those wishing to overtake, but unable to do so.

**Table B.1:** Number and percentage of overtakes occurred and attempted. Note the diversity in intent and in overtaking-difficulty for the subjects, motivating the need for assistive shared autonomy.

| Overtake occurred? | Frequency | Percentage | Overtake attempted? | Frequency | Percentage |
|---|---|---|---|---|---|
| "No" | 178 | 30.07 | "No" | 65 | 11.02 |
| "Yes" | 414 | 69.93 | "Yes" | 525 | 88.98 |

We consider additional statistics, including statistics on left- and right-hand passing, as well as collisions with the other vehicle or objects, and spin-outs. We include these results in Table B.2. We note that there is a nearly-equal number of overtakes on the right versus left. On an individual level, we ran chi-square tests of test for given probabilities to look for side preferences. We found that only 6 of the 48 participants showed a statistically significant ($p < .05$) passing side bias, with two participants having a bias for the right side and four participants having a bias for the left side B.2. We also note that participants were, in general, imperfect in their driving, with nearly 50% of trials having a collision and 8% having at least one spin-out.

## C  Human Objectives

In this section, we discuss the reward terms used to generate the explicit decision-making tendencies of the fictitious human drivers, via $r_t^H = r_t^{task} + r_t^y$. Each use the task-specific reward terms outlined

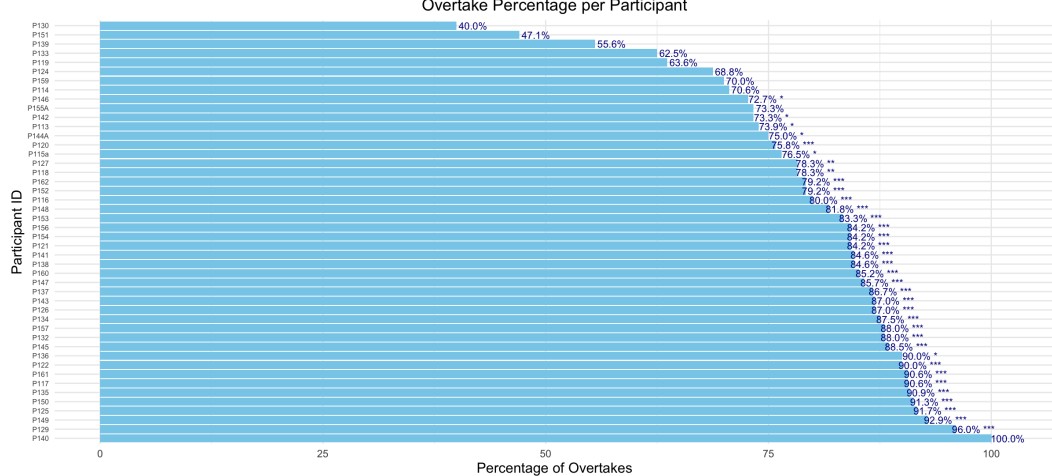

**Figure B.1:** Consistency of overtake versus non-overtakes.

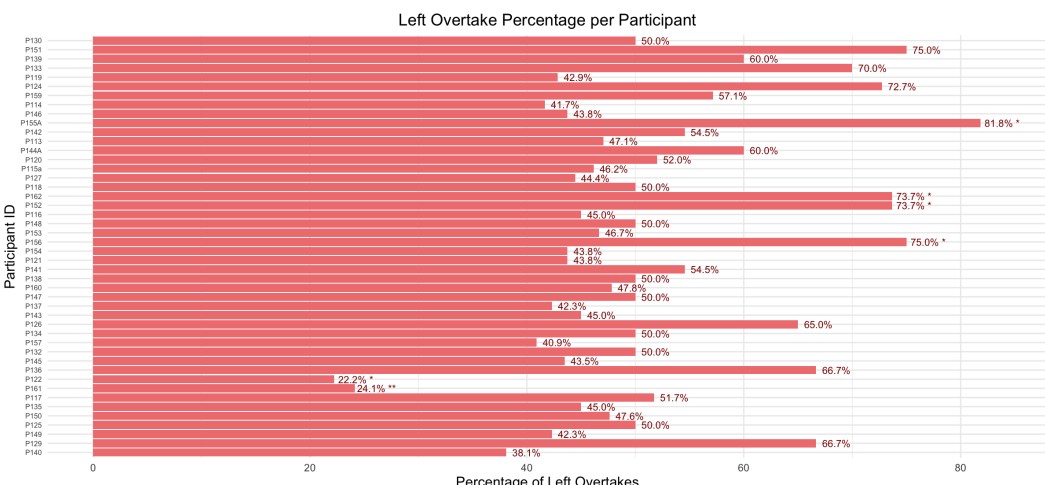

**Figure B.2:** Consistency of left-handed versus right-handed overtakes.

in (1), in combination with objective-specific rewards. Many of the task rewards are borrowed from [15]. We focus here on the human objective term $r_t^y$.

**Pass** We adopt a dense reward that provides a penalty when the vehicle is behind the opponent vehicle, and a reward bonus when in front of that vehicle, up to a threshold, to incentivize passing. That is,

$$r_t^y = c_{pass} \left(\Delta s_t - \Delta s_{t-1}\right) \mathbb{I}\left((s_{low} \leq \Delta s_t \leq s_{high}) \vee (s_{low} \leq \Delta s_{t-1} \leq s_{high})\right)$$

Where $\mathbb{I}(\cdot)$ is the indicator function, and $\Delta s_t$ is the difference in longitudinal positions, relative to track coordinates, between the ego and opponent vehicles, $\Delta s_t = s_t^{ego} - s_t^{opp}$. We take the scalar $c_{pass} = 10$. In other words, if the difference between $s_t^{ego}$ (the ego position) and $s_t^{opp}$ (the opponent position) is between $s_{low}$ and $s_{high}$, the passing reward is equal to $10 * \left(s_t^{ego} - s_t^{opp}\right)$. This means there is a high positive reward for getting far ahead of the opponent, and a high negative reward for falling behind the opponent. For both the *pass* reward, we set $s_{high} = -s_{low} = 800$, which ensures that the pass reward is active for the entire trial. Note that we do not impose a progress reward with the pass objective.

**Stay-Behind** Due to the non-symmetry of the problem, the stay-behind reward cannot be the complement of the pass reward (otherwise, the stay-behind agent would drive backwards to get away

**Table B.2:** Other effects. Percentages are percent of trials with the listed event.

| Observed event | Percentage |
|---|---|
| Left-handed overtakes | 50.82% |
| Right-handed overtakes | 49.18% |
| Collisions | 48.66% |
| Spin-outs | 8.36% |

from the opponent). Because the task reward does not consider making progress, we add that here to the human-specific reward. The stay-behind reward is then:

$$r_t^y = r_t^{prog} + c_{stay} (\Delta s_t - \Delta s_{t-1}) \mathbb{I} ((s_{low} \le \Delta s_t \le s_{high}) \vee (s_{low} \le \Delta s_{t-1} \le s_{high}))$$

Where $c_{stay} = -2$, and we impose a progress reward similar to [15], i.e., $r_t^{prog} = s_t^{ego} - s_{t-1}^{ego}$, with $s_t^{ego}$ being the ego's longitudinal position in track coordinates. For the *stay-behind* reward, we set $s_{high} = -s_{low} = 50$. In practice, this means that the stay-behind agent is encouraged to make progress along the track ($r^{prog}$), but to stay at least 50 meters behind the opponent.

Both the left- and right-biased passing agents are passing agents with an additional reward term that encourages a bias to the left or right. Note that these additional treatments do not guarantee passing on one side or the other.

**Left-Biased** We adopt a reward bonus for driving on the opponent's left; i.e.

$$r_t^y = (\Delta s_t - \Delta s_{t-1}) \mathbb{I} ((s_{low} \le \Delta s_t \le s_{high}) \vee (s_{low} \le \Delta s_{t-1} \le s_{high})) + (\Delta e_t + c_{margin})$$

where $\Delta e_t$ is the difference in lateral positions of the two vehicles, in the track coordinate frame; i.e. $\Delta e_t = e_t^{ego} - e_t^{opp}$, and $c_{margin}$ is a margin (which we set to $c_{margin} = 0.3$).

**Right-Biased** Right-biased reward is the complement of the left-biased reward:

$$r_t^y = (\Delta s_t - \Delta s_{t-1}) \mathbb{I} ((s_{low} \le \Delta s_t \le s_{high}) \vee (s_{low} \le \Delta s_{t-1} \le s_{high})) - (\Delta e_t + c_{margin})$$

## D    Additional Model Details

We provide a summary of the DREAM2ASSIST training procedure in Algorithm 1. The procedure is split into two phases: the first is a human population generation phase in which we use the rewards in Sec. C to generate a population of humans included in the tuple $\langle \mathcal{W}_i^H, \Pi_i^H \rangle$ of world models and policies, respectively, and the expert human models denoted by the tuple $\langle W_i^*, \pi_i^* \rangle$ for each human objective $y_i$. The second phase entails drawing from samples of $\langle \{y_j\}_{j=0}^N, \{\mathcal{W}_j^H\}_{j=0}^N, \{\Pi_j^H\}_{j=0}^N \rangle$ using fictitious co-play (FCP) [27] in order to train the assistant's world model $W^A$ and policy $\pi^A$. At runtime, both the trained policy $\pi^A$ and world model $W^A$ are executed, with $W^A$ being additionally useful as a means to interpret the decisions made by $\pi^A$; e.g. the intent estimate $\hat{y}_t^A$, the estimated reward $\hat{r}_t^A$ or the latent variables $\hat{z}_t$.

**Table C.1:** Training Hyperparameters.

| Hyperparameter | Value | Hyperparameter | Value |
|---|---|---|---|
| Encoder / decoder MLP layers | 2 | Steps | 2e6 |
| Encoder / decoder MLP units | 512 | Batch size | 16 |
| Predictor head layers | 2 | Batch length | 64 |
| Predictor head units | 512 | Training ratio | 512 |
| Discount factor | 0.997 | Model learning rate | 1e-4 |
| Discount $\lambda$ | 0.95 | Value learning rate | 3e-5 |
| Imagined horizon | 15 | Actor learning rate | 3e-5 |
| Actor entropy | 3e-4 | Dataset max size | 1e6 |
| Dynamics hidden units | 512 | # Steps between evaluations | 1e4 |
| Dynamics discrete dimension | 32 | # Episodes to evaluate | 10 |

---
**Algorithm 1** DREAM2ASSIST using FCP
---
**Given**: diverse intents, $y_i \in \{1, 2, \ldots, M\}$
**Given**: reward functions for each intent $y$
**for** $y_i, i \in \{1, 2, \ldots, M\}$ **do**                           ▷ Generate human population
    Initialize $\pi_i^H, W_i^H$
    **while** not converged **do**
        Sample an opponent policy $\pi_{opp}$ from $\Pi_{opp}$
        Initialize $\boldsymbol{x}_0$, with $t = 0$
        **while** not done **do**
            Perform gradient step; update $\pi_i^H, W_i^H$
            Sample action $\boldsymbol{a}_{i,t}^H$ from $\pi_i^H$
            Step the environment with action $\boldsymbol{a}_{i,t}^H$ and $\boldsymbol{a}_t^{opp} \sim \pi^{opp}$
            Shape rewards according to $i$th agent reward
            $t \leftarrow t + 1$
        **end while**
        Append checkpoint $\langle W_i^H, \pi_i^H \rangle$ to $\langle \mathcal{W}_i^H, \Pi_i^H \rangle$
    **end while**
    Append final $\langle W_i^*, \pi_i^* \rangle$ to $\langle \mathcal{W}_i^H, \Pi_i^H \rangle$
**end for**
Freeze agents and world models $\{\mathcal{W}_j^H\}_{j=0}^N, \{\Pi_j^H\}_{j=0}^N$
Initialize $\pi^A, W^A$
**while** not converged **do**                                     ▷ Train assistant agent
    Sample intent $i$ from $\langle \{y_i\}_{i=0}^M \rangle$
    Sample checkpoint $j$ for intent $y_i$ from $\langle \{\mathcal{W}_j^H\}_{j=0}^N, \{\Pi_j^H\}_{j=0}^N \rangle$
    Sample an opponent policy $\pi_{opp}$ from $\Pi_{opp}$
    Initialize $\boldsymbol{x}_0$, with $t = 0$
    **while** not done **do**
        Perform gradient step; update $\pi^A, W^A$ using ground truth label $y_i$
        Sample action $\boldsymbol{a}_t^A$ from $\pi^A(\boldsymbol{x})$, $\boldsymbol{a}_t^H$ from $\pi_j^H(\boldsymbol{x}_t)$
        Step the environment using shared action $\boldsymbol{a}_t^H + \boldsymbol{a}_t^A$ and $\boldsymbol{a}_t^{opp} \sim \pi^{opp}$
        Evaluate intent $\hat{y}_t^A$ using $W^A$
        Shape rewards as per (6), using $r^*(\hat{y}_t^A)$ from $W_j^*$, $\boldsymbol{a}_t^*(\hat{y}^A) \sim \pi_{\hat{y}_t^A}^*(\boldsymbol{x}_t)$
        $t \leftarrow t + 1$
    **end while**
**end while**
---

## D.1 Training Hyperparameters and Environment Specifics

We provide the DREAM2ASSIST hyperparameters in Table C.1. We train using the Adam optimizer for $2 \times 10^6$ steps.

The CARLA simulator is used for our environment, and is executed with step size of 0.1 sec. We terminate episodes if: (a) the ego collides with the opponent or other collidable objects (e.g. static barriers), (b) the ego vehicle veers too far off course, or (c) a predefined finish line is reached. The map used is a geospatially-calibrated representation of the Thunderhill Raceway in Willows, CA.

## E Additional Experimental Results

### E.1 Summarized Performance

We summarize the discussion in Sec. 4.2 and 4.1 here in Fig. E.1.

### E.2 Performance Across Different Humans

We provide a more complete comparison of the results, showing additional metrics in an evaluation that extends Fig. 3 from imperfect to near-perfect humans (1–11, ordered according to unassisted track progress performance) for **DREAMER**, **DREAMER-AIL**, **BETAIL**, and **DREAM2ASSIST**. We show these results in Figs. E.2–E.5. Note that the additional fictitious humans (6–11) achieve

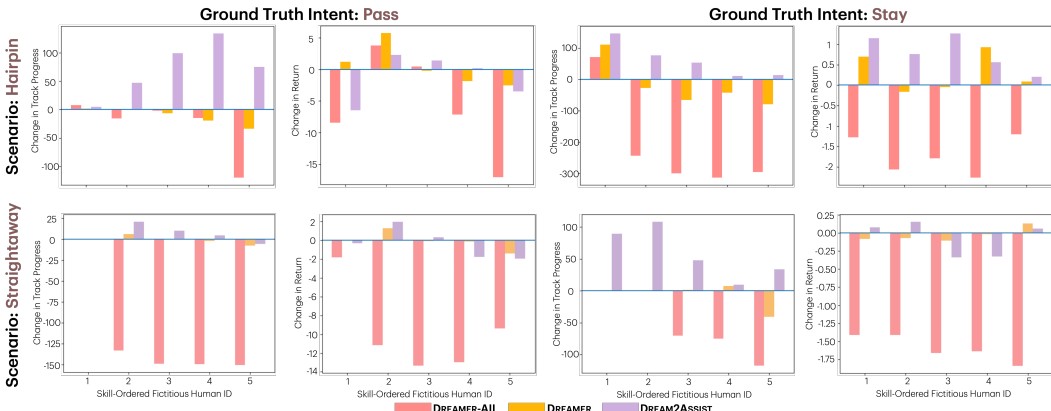

**Figure E.1:** Change in track progress and return when adding assistance to five imperfect pass (left) and stay (right) fictitious humans in the hairpin and straightaway problem settings. The addition of DREAM2ASSIST leads to higher gains in task performance and greater adherence to human objectives than baselines.

nearly-identical baseline performance across all metrics, as indicated by Figs. E.3 and E.5. Hence, the changes across humans 6–11 in Figs. E.2 and E.4 are likewise similar.

From Fig. E.2, we observe that DREAM2ASSIST, when applied to the imperfect *pass* humans (1–5) generally yield improvements in progress, collisions, and speed, with a slight overall decrease in reward, and an overall moderate intervention level compared to the baselines. For humans 6–10, progress, reward, collisions, and speed are all negatively impacted, hinting at room for improvement in handling near-perfect humans. Similar trends for imperfect humans (1–5) tending to *stay* can be seen in Fig. E.4, where all metrics except collision see improvement. For near-perfect humans (6–11), the results generally indicate marginal improvement across all metrics, except collision.

We also complement Table 1 with additional metrics, including the magnitude of intervention, and speed, both averaged over time. In Tables E.1 and E.2, we compare **Dreamer**, **DREAM2ASSIST**, and **DREAM2ASSIST-AIL**, and further include results for a variant of DREAM2ASSIST with the action-based reward term in (6) using $\alpha_r = \alpha_a = 1$, which we call **DREAM2ASSIST+a**, as well as a variant of DREAM2ASSIST with the action-based reward term and no reward, i.e. $\alpha_a = 1$, $\alpha_r = 0$, which we call **DREAM2ASSIST+a−r**. For the pass vs. stay case in Table E.1, DREAM2ASSIST achieves best performance in 5 categories, while improving over unassisted humans in 10 categories. The performance of DREAM2ASSIST+a and DREAM2ASSIST+a−r were mixed. DREAM2ASSIST+a was able to improve over unassisted humans, but with generally lower progress than DREAM2ASSIST, while DREAM2ASSIST+a−r almost completely hindered the human's progress, due to the fact that the reward term no longer explicitly captures the dense progress sub-reward, and are not implicitly reflected in the actions of the optimal human. In E.2, we see similar trends, with DREAM2ASSIST outperforming baseline approaches in 5 categories, and performing better than unassisted humans in 8 categories. DREAM2ASSIST+a−r is unable to make progress, and DREAM2ASSIST+a also reveals lower progress than the DREAM2ASSIST in the hairpin and straightaway domains.

### E.3 Intent Classification Performance

We next probe the performance of the intent classification. $F_1$ scores achieved on training data yields high performance, as shown in Table E.3.

We provide two example time traces to illustrate stability of inferring the human's intent by the assistant's world model in Figs. E.7 and E.8.

To uncover whether intent inference is due to world model training, we evaluate the t-SNE embeddings of the logits of the assistant's discrete latent state $\hat{z}_t^A$. We see that in the Dreamer case, t-SNE is unable to find strong separations between the ground truth intent classes without intent inference in the latents, while in DREAM2ASSIST, there is a stronger separation, and the ground truth intent

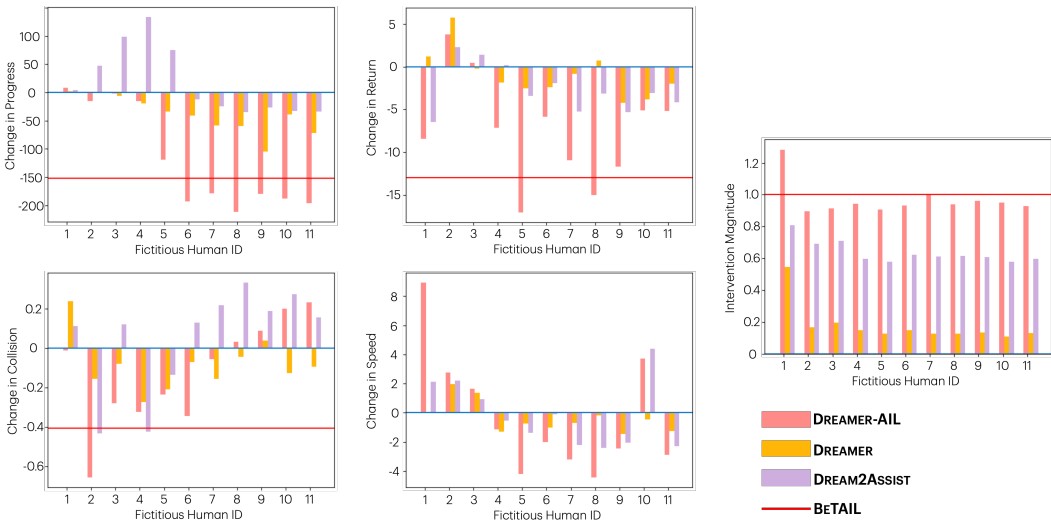

**Figure E.2:** Changes in various metrics in the hairpin scenario when adding assistance to various imperfect (1–5) and near-perfect (6–11) humans tending to *pass*, averaged over four random seeds. Due to the fact that BeTAIL uses its own internal human model, we compare only one instance / human, as denoted by the red line.

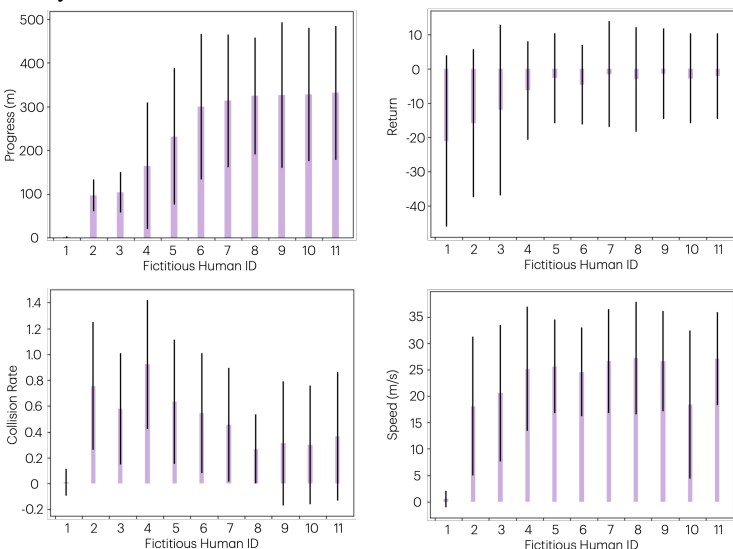

**Figure E.3:** Absolute metrics in the hairpin scenario evaluated for the *unassisted* imperfect (1–5) and near-perfect (6–11) humans tending to *pass*, averaged over four random seeds. 1-$\sigma$ error bars are shown.

classes are more strongly clustered in the embedding space, allowing intent to be inferred with much higher accuracy.

## E.4 Learning Curves

We provide learning curves for the **DREAMER**, **DREAM2ASSIST** and **DREAM2ASSIST**+**a** assistance schemes in the two domains across all the human objectives in Fig. E.9. We compare these across track progress, and observe a general trend of stability in training.

## E.5 Visuals of Assistance

We show, in Fig. E.11, CARLA and bird's-eye-view snapshots of driving with assistance, and compare that to a human without assistance in Fig. E.10.

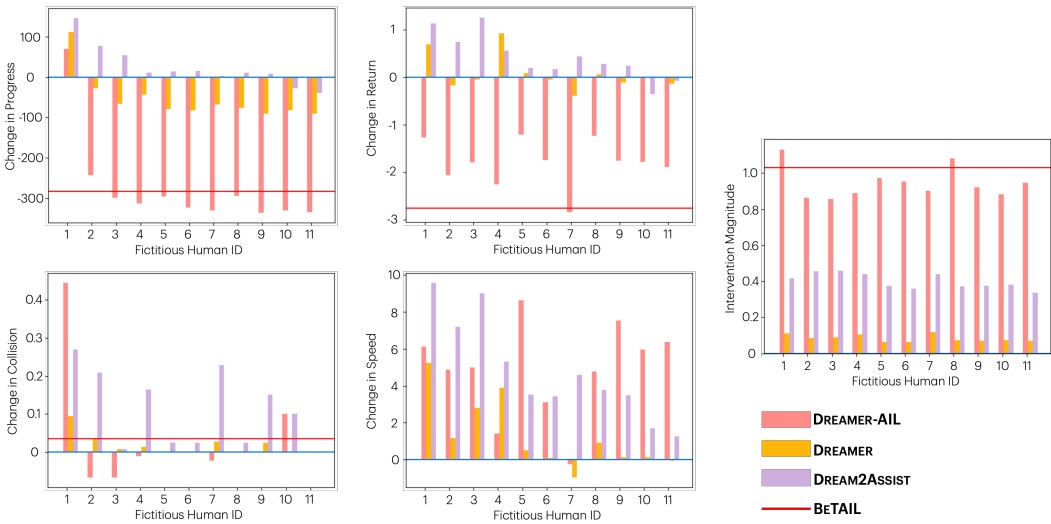

**Figure E.4:** Changes in various metrics in the hairpin scenario when adding assistance to various imperfect (1–5) and near-perfect (6–11) humans tending to *stay*, averaged over four random seeds. Due to the fact that BeTAIL uses its own internal human model, we compare only one instance / human, as denoted by the red line.

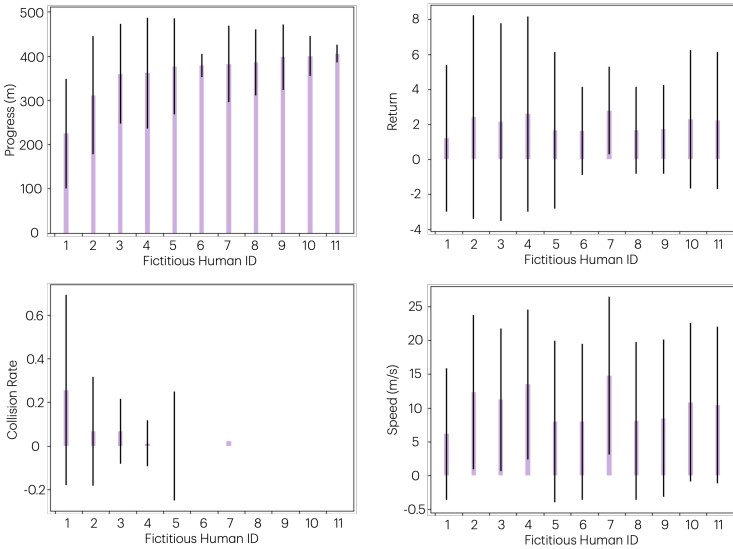

**Figure E.5:** Absolute metrics in the hairpin scenario evaluated for the *unassisted* imperfect (1–5) and near-perfect (6–11) humans tending to *stay*, averaged over four random seeds. 1-$\sigma$ error bars are shown.

## F  Comparisons Without a Recurrent State Space Model

For completeness, we compare to a simple windowed-observation baseline agent that does not use a recurrent state space model, yet still learns to perform intent classification. We keep our training pipeline and synthetic human partners the same as that used for DREAM2ASSIST for these experiments. This agent (NO-RSSM) is given 16 frames of prior history (corresponding to about 1.5 seconds of observations) and is trained with the same rewards as DREAM2ASSIST, in addition to an auxiliary intent-classification objective. We train this agent to assist *pass* and *stay-behind* human partners on the hairpin section of the track.

The results of this experiment are in Figures F.1 & F.2. We observe that the NO-RSSM baseline performs significantly worse than DREAM2ASSIST and worse than most RSSM-based methods, despite having high intent classification accuracy (98% F1, equal to the DREAM2ASSIST agent during

**Table E.1:** Improvement over unassisted humans for the *pass–stay* humans on straightaway and hairpin experiments, with statistics aggregated across four random seeds. Blue indicates improvement over unassisted humans, **bold** is best.

| | Pass (top) / Stay (bottom) | | | | | | | | | |
|---|---|---|---|---|---|---|---|---|---|---|
| | | | Hairpin | | | | | Straightaway | | |
| | Progress (m)↑ | Return↑ | Collisions↓ | Interventions↓ | Speed (m/s)↑ | Progress (m)↑ | Return↑ | Collisions↓ | Interventions↓ | Speed (m/s)↑ |
| DREAMER | -11.3± 13.6 | **0.5± 2.9** | -0.1± 0.2 | 0.2 ± 0.2 | 0.1 ± 1.1 | -0.7± 4.6 | **-0.1± 0.9** | 0.0± 0.1 | 0.1 ± 0.0 | **-0.4 ± 1.9** |
| | -21.1± 68.6 | 0.3± 0.4 | 0.0± 0.0 | 0.1 ± 0.0 | 2.9 ± 1.6 | -6.1± 17.5 | **0.0± 0.1** | 0.1± 0.2 | 0.1 ± 0.0 | 0.3 ± 0.5 |
| DREAMER-AIL | -28.9± 46.2 | -5.7± 7.3 | **-0.3± 0.2** | 1.0 ± 0.2 | **1.4 ± 4.4** | -116.6± 58.7 | -9.7± 4.2 | **-0.4± 0.3** | 1.4 ± 0.0 | -21.2 ± 10.6 |
| | -216.6± 145.6 | -1.7± 0.4 | 0.1± 0.2 | 0.9 ± 0.1 | 5.4 ± 2.4 | -52.5± 45.9 | -1.6± 0.2 | **-0.1± 0.1** | 1.4 ± 0.0 | -1.2 ± 1.1 |
| DREAM2ASSIST | **71.8± 43.9** | -1.2± 3.3 | -0.2± 0.2 | 0.7 ± 0.1 | 0.5 ± 1.4 | 6.0± 9.1 | -0.4± 1.4 | -0.1± 0.1 | 0.3 ± 0.0 | -1.2 ± 1.9 |
| | **60.5± 49.7** | **0.8± 0.4** | 0.1 ± 0.1 | 0.4 ± 0.0 | **7.1 ± 2.4** | **57.8± 36.4** | -0.1± 0.2 | 0.1 ± 0.1 | 0.4 ± 0.1 | 1.7 ± 1.0 |
| DREAM2ASSIST+a | 25.2 ± 11.0 | -2.1 ± 7.4 | -0.3 ± 0.2 | 0.7 ± 0.2 | 0.8 ± 4.9 | 6.9 ± 9.9 | -5.6 ± 2.4 | -0.1± 0.2 | 0.8 ± 0.2 | -14.8 ± 7.7 |
| | 24.3 ± 66.6 | 0.2 ± 0.6 | 0.0 ± 0.4 | 0.7 ± 0.0 | 5.1 ± 3.0 | 9.2 ± 7.8 | -6.6 ± 0.7 | 0.1 ± 0.1 | 0.9 ± 0.1 | 0.4 ± 0.2 |
| DREAM2ASSIST+a−r | -79.0 ± 53.2 | -13.5 ± 3.9 | -0.5 ± 0.6 | 0.6 ± 0.0 | -6.1 ± 5.6 | -9.0 ± 20.2 | -2.9 ± 1.0 | 0.0 ± 0.1 | 0.3 ± 0.3 | -0.7 ± 1.2 |
| | -281.9 ± 68.7 | -2.5 ± 0.5 | **-0.1 ± 0.1** | 0.7 ± 0.0 | -1.4 ± 2.3 | 10.4 ± 38.3 | -1.0 ± 0.1 | 0.1 ± 0.3 | 0.8 ± 0.2 | **1.8 ± 0.5** |

training). While the NO-RSSM agent is trained with the same rewards as the DREAM2ASSIST agent, it regularly terminates episodes early by steering the human driver out of bounds, which could be the result of a poor understanding of when it will receive positive or negative rewards (as the two intents result in opposite reward signals). As the NO-RSSM agent is not trained to predict reward, this information may not be as explicitly captured in its internal representation as it would be in an RSSM based agent, such as DREAMER or DREAM2ASSIST.

# G   Evaluations With Random Intent Transitions

While our primary experiments focus on humans with a static intent (i.e., the human is attempting to *pass* or *stay-behind* for the entire episode), here we include additional experiments in which our synthetic humans randomly change intents in the middle of an episode. For these experiments, we run 10 trials and, after a randomly sampled number of episode steps, we swap out the synthetic human for a comparably-trained synthetic human with the *opposite* intent. For example, we might start an evaluation trial with a *pass* human that is trained for 10K steps and, after 10 seconds of the trial, suddenly swap the synthetic-human policy to a *stay-behind* human that has been trained for 10K steps.

We run these experiments with a standard DREAMER baseline and our DREAM2ASSIST agent, evaluating the impact of our inferred-reward and intent classification objective. These experiments are repeated with 5 random seeds for a total of 50 episodes for each randomized transition. Random seeds are kept consistent for the DREAMER and DREAM2ASSIST experiments, so the 50 randomly-selected transition times are consistent for the two agents.

We note that our assistants have never been trained under such dynamics, and so these transitions are highly out-of-distribution for both the assistants *and* the synthetic humans. Our synthetic human agents are therefore very likely to crash, spin-out, or otherwise perform far worse than usual. To evaluate our assistive agents in this domain, we compare the collision rate, average change in speed, average track progress (i.e., task-completion), and intervention norm from the assistive agents for randomized pass-to-stay transitions (Figure G.1) and randomized stay-to-pass transitions (Figure G.2).

While the DREAMER and DREAM2ASSIST agents can both help to mitigate the problems inherent to a random intent transition (collisions and spin-outs), we observe that DREAM2ASSIST offers

**Table E.2:** Improvement over unassisted humans for the *left–right* humans on straightaway and hairpin experiments, with statistics aggregated across four random seeds. Blue indicates improvement over unassisted humans, **bold** is best.

| | Left (top) / Right (bottom) | | | | | | | | | |
|---|---|---|---|---|---|---|---|---|---|---|
| | | | Hairpin | | | | | Straightaway | | |
| | Progress (m)↑ | Return↑ | Collisions↓ | Interventions↓ | Speed (m/s)↑ | Progress (m)↑ | Return↑ | Collisions↓ | Interventions↓ | Speed (m/s)↑ |
| DREAMER | 21.8± 28.7 | -2.0± 2.2 | 0.1± 0.1 | 0.14 ± 0.1 | 0.4 ± 1.2 | **10.8± 7.3** | -0.6± 0.3 | 0.0± 0.1 | 0.2 ± 0.0 | **6.0 ± 7.0** |
| | 10.0± 17.9 | -1.3± 1.8 | 0.0± 0.1 | 0.2 ± 0.1 | 0.1 ± 1.5 | **-1.1± 6.2** | **0.2± 1.2** | 0.0± 0.1 | 0.1 ± 0.0 | **0.9 ± 1.0** |
| DREAMER-AIL | -144.0± 89.9 | -7.1± 6.6 | **-0.4± 0.2** | 1.2 ± 0.1 | -14.4 ± 9.5 | -134.4± 13.0 | -2.5± 0.9 | **-0.5± 0.1** | 1.4 ± 0.0 | -11.4 ± 7.8 |
| | -119.1± 79.9 | -4.5± 17.3 | **-0.4± 0.3** | 1.3 ± 0.0 | -15.2 ± 5.6 | -126.3± 53.2 | -13.4± 7.0 | **-0.5± 0.2** | 1.4 ± 0.0 | -23.1 ± 5.1 |
| DREAM2ASSIST | **54.8± 60.4** | 1.4± 5.2 | 0.0± 0.1 | 0.8 ± 0.1 | -0.6 ± 6.0 | 5.0± 5.0 | 0.1± 0.5 | 0.0± 0.1 | 0.6 ± 0.1 | -0.1 ± 1.2 |
| | **27.2± 23.1** | **2.2± 2.3** | -0.2± 0.2 | 0.8 ± 0.1 | **3.2 ± 4.7** | -1.1± 31.9 | -2.8± 2.4 | 0.2± 0.2 | 0.8 ± 0.1 | -2.1 ± 4.3 |
| DREAM2ASSIST+a | 9.3 ± 32.0 | **3.2 ± 4.3** | -0.1 ± 0.1 | 0.9 ± 0.1 | **2.9 ± 5.8** | -18.3 ± 12.9 | 0.2 ± 1.7 | -0.2 ± 0.2 | 0.9 ± 0.1 | -4.9 ± 5.3 |
| | -61.2 ± 40.5 | -0.2 ± 2.1 | -0.2 ± 0.3 | 0.9 ± 0.1 | -1.4 ± 3.1 | -8.2 ± 35.6 | -5.2 ± 5.4 | -0.1 ± 0.2 | 1.0 ± 0.0 | -14.0 ± 6.0 |
| DREAM2ASSIST+a−r | -147.3 ± 93.0 | -24.2 ± 13.1 | **-0.4 ± 0.2** | 0.5 ± 0.1 | -9.6 ± 7.7 | 6.9 ± 4.1 | **0.3 ± 1.5** | 0.0 ± 0.1 | 0.4 ± 0.1 | **5.2 ± 3.9** |
| | -179.9 ± 104.2 | -22.2 ± 16.3 | **-0.4 ± 0.4** | 0.6 ± 0.1 | -10.7 ± 6.8 | -3.2 ± 12.6 | -0.0 ± 3.0 | 0.1 ± 0.2 | 0.1 ± 0.0 | -0.1 ± 1.7 |

**Table E.3:** $F_1$-Scores over the training set.

| Pass vs. Stay Hairpin | Pass vs. Stay Straightaway | Left vs. Right Hairpin | Left vs. Right Straightaway |
|:---:|:---:|:---:|:---:|
| $0.99 \pm 0.006$ | $1.00 \pm 0.000$ | $0.95 \pm 0.00$ | $0.98 \pm 0.00$ |

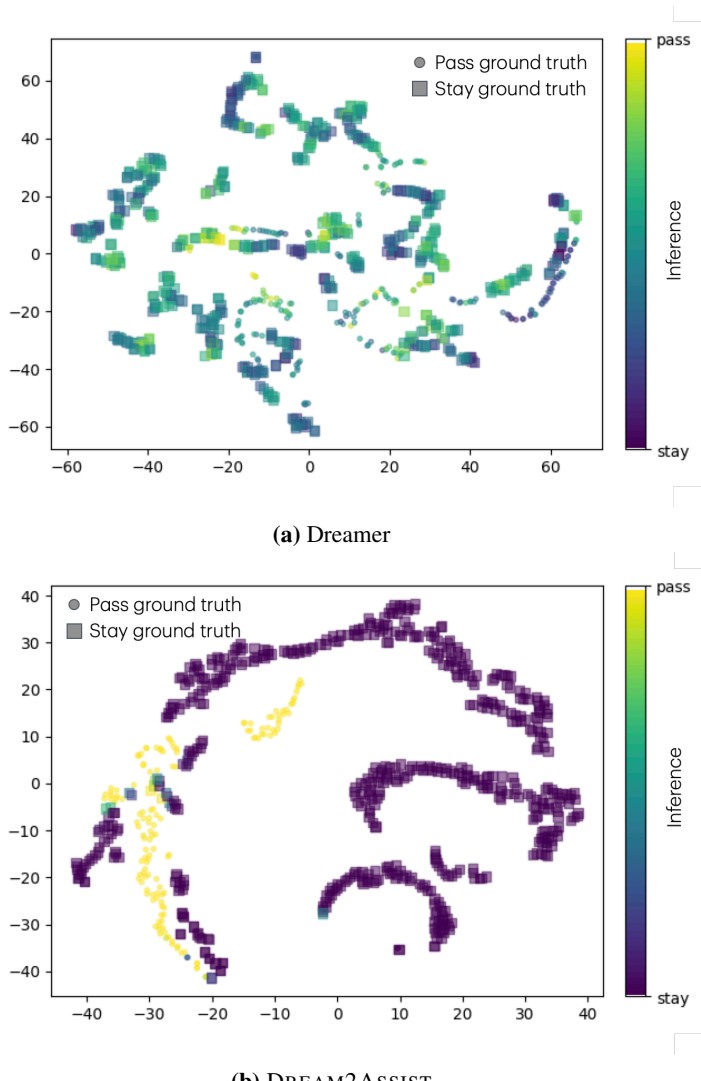

**(a)** Dreamer

**(b)** DREAM2ASSIST

**Figure E.6:** t-SNE embeddings of $\hat{z}_t^A$ for: (a) the non-intent-aware world model of Dreamer versus (b) the intent-supervised world model of DREAM2ASSIST. The consistency of the clusterings present in the DREAM2ASSIST world model states indicates that the world model has learned to identify the human's intent.

a much greater benefit than the standard DREAMER agent. Immediately after a random intent transition, the human agents perform very poorly and need significant assistance to get "back on track" to their known state distributions. DREAM2ASSIST consistently outperforms DREAMER for worse-performing human partners, as the intervention magnitudes for the DREAM2ASSIST agent consistently show more intervention and assistance. While the DREAMER agent pairs well with highly performant human partners (due to the low intervention magnitude), DREAMER is not able to offer as much useful assistance in this experimental setting.

Finally, we show the intent-classification accuracy during a stay-to-pass transition episode with a medium-performance synthetic human (Figure G.3), and during a pass-to-stay transition episode with

a high-performance synthetic human (Figure G.4). In these figures, we see that intent-classification accuracy is quite high before the random transition ($> 90\%$), and drops sharply when the transition occurs (as expected). In the seconds that follow, the agent slowly recovers the new ground-truth intent as it observes different behavior from the human in the environment. Within 4-8 seconds after the random transition, we observe that the DREAM2ASSIST agent has recovered $> 90\%$ intent-classification performance with the new human partner.

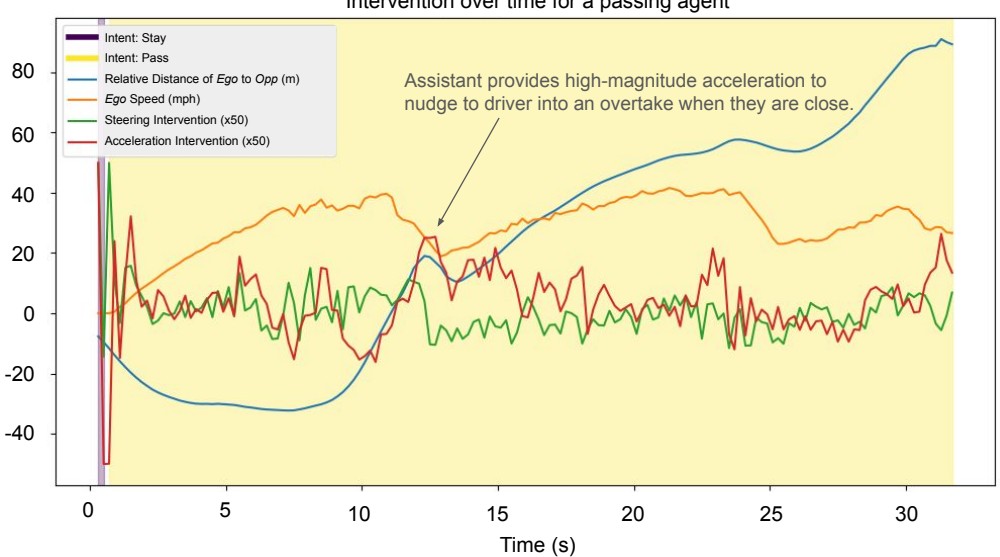

**Figure E.7:** Time traces in the hairpin scenario showing the intent inference (denoted by background color), along with ego-opponent distance, speed, and DREAM2ASSIST steering and acceleration modifications for a human tending to *pass*. Notice that DREAM2ASSIST maintains an accurate estimate of the driver's intent, and provides a high-magnitude acceleration intervention to assist as the ego begins to overtake the opponent.

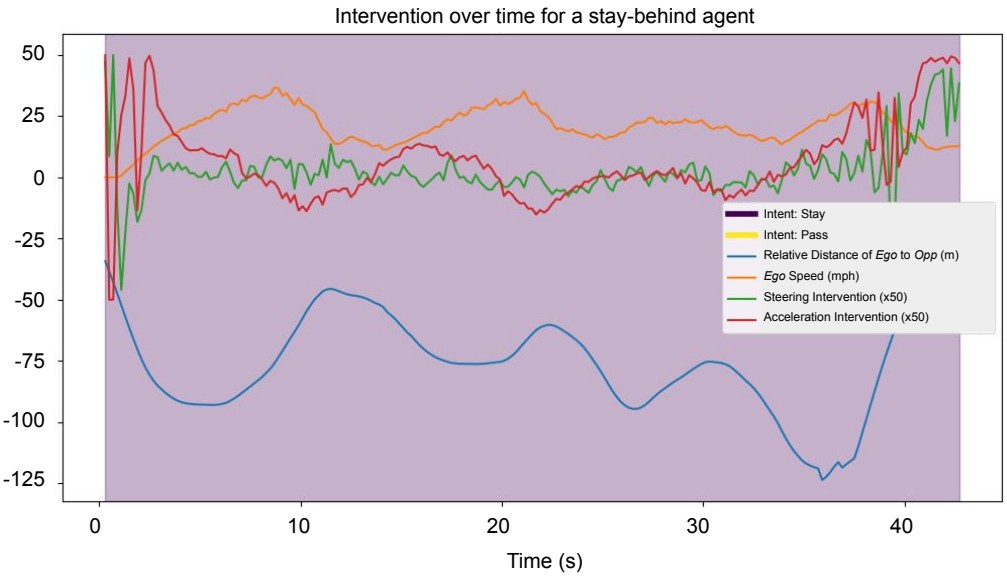

**Figure E.8:** Time traces in the hairpin scenario showing the intent inference (denoted by background color), along with ego-opponent distance, speed, and DREAM2ASSIST steering and acceleration modifications for a human tending to *stay*.

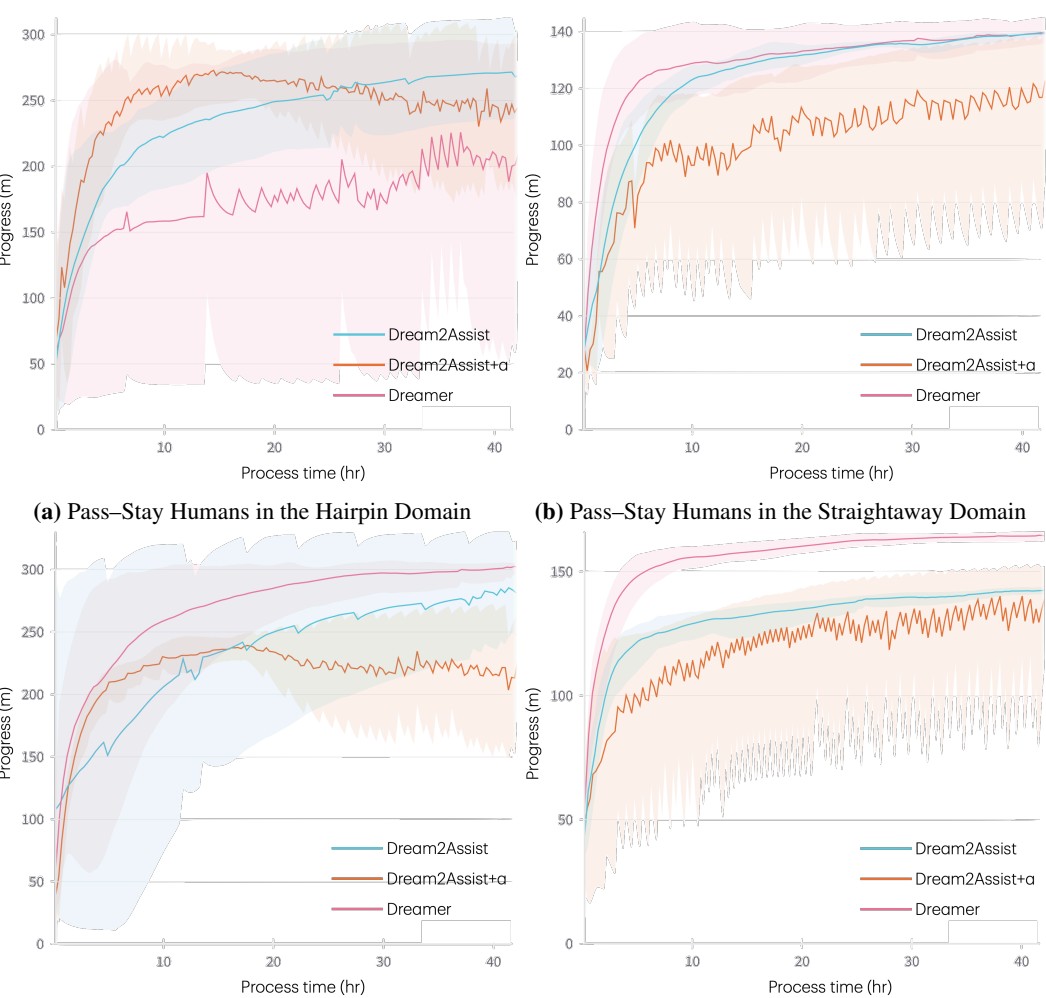

**(a)** Pass–Stay Humans in the Hairpin Domain

**(b)** Pass–Stay Humans in the Straightaway Domain

**(c)** Left–Right Humans in the Hairpin Domain.

**(d)** Left–Right Humans in the Straightaway Domain.

**Figure E.9:** Learning curves for each domain, averaged over four random seeds.

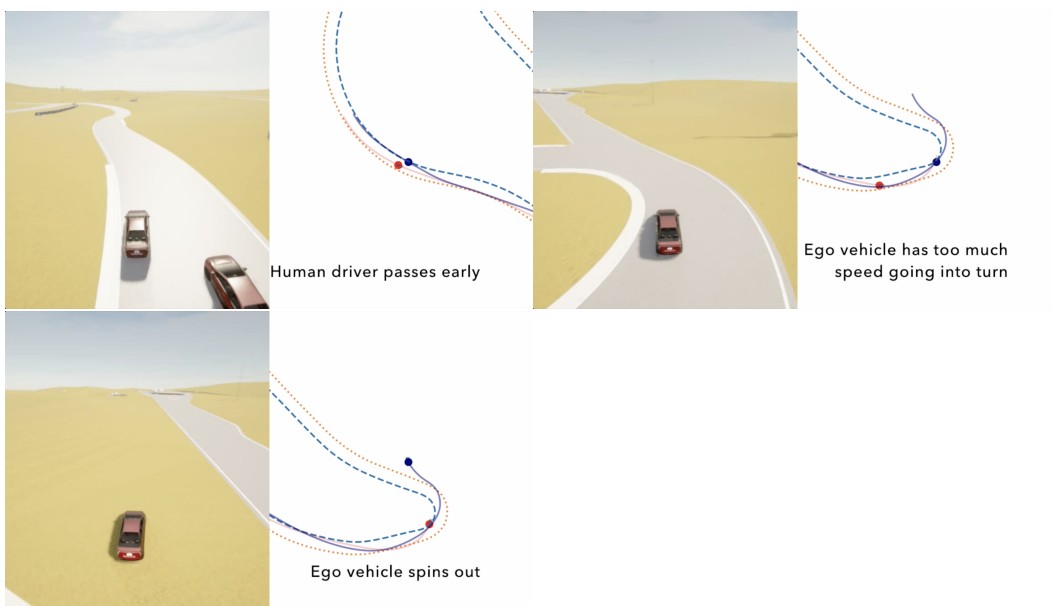

**Figure E.10:** Example of a time sequence of an imperfect passing human driving in the hairpin domain.

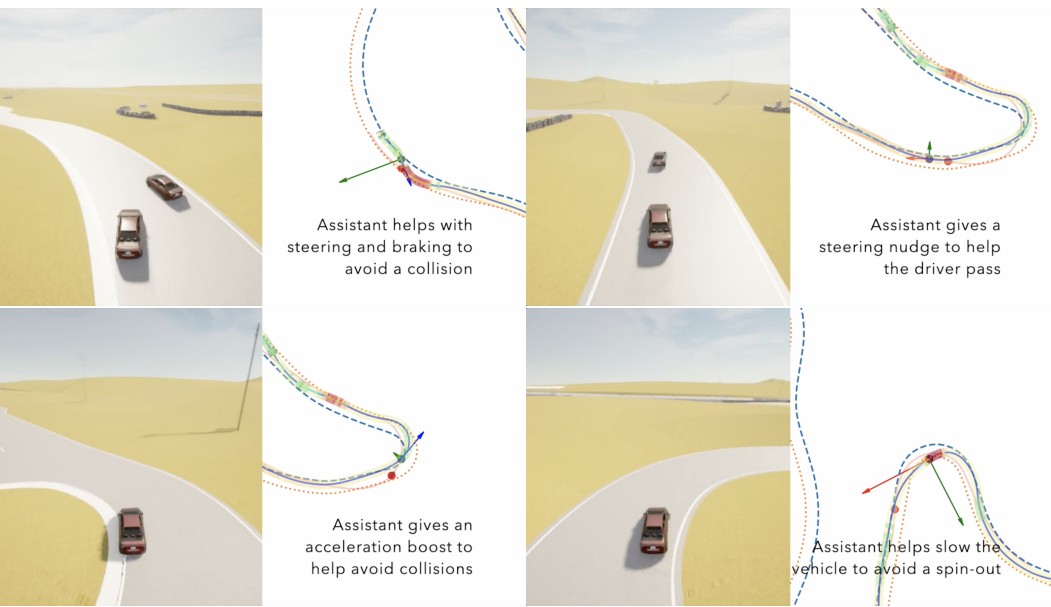

**Figure E.11:** Example of a time sequence of DREAM2ASSIST assistance to help an imperfect passing human in the hairpin domain.

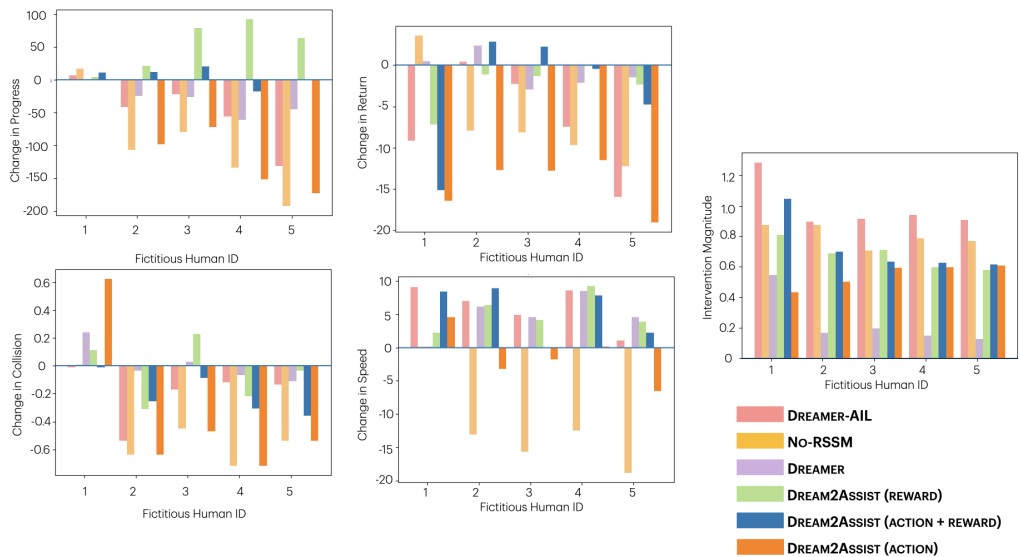

**Figure F.1:** Changes in various metrics in the hairpin scenario when adding assistance to various imperfect (1–5) humans tending to *pass*, averaged over five random seeds. Note that our two NO-RSSM baseline is trained to perform intent classification and achieves high accuracy, yet performs far worse when paired with sub-optimal synthetic human partners than our RSSM baselines.

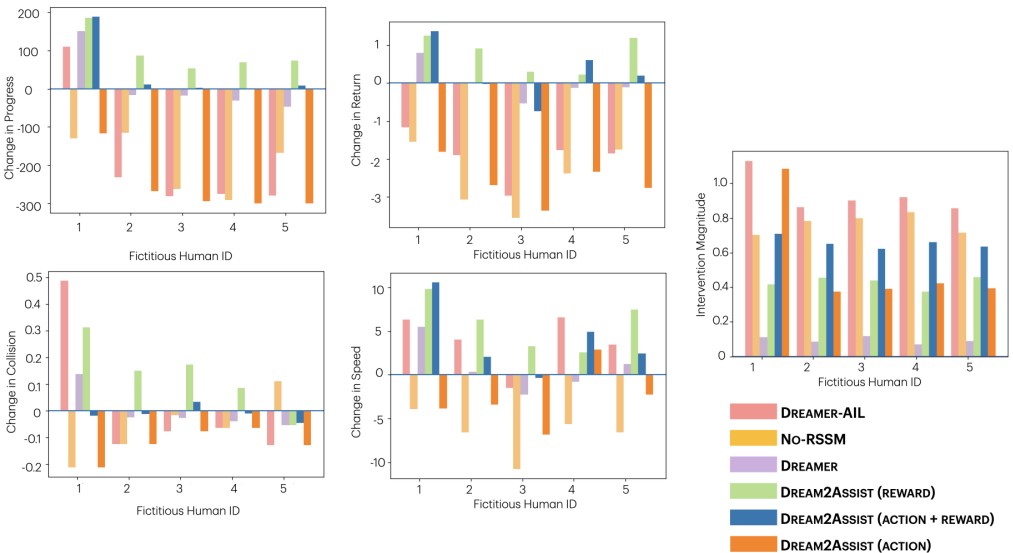

**Figure F.2:** Changes in various metrics in the hairpin scenario when adding assistance to various imperfect (1–5) humans tending to *stay-behind*, averaged over five random seeds. We again note that the NO-RSSM baseline is trained to perform intent classification and achieves high accuracy, yet performs far worse when paired with sub-optimal synthetic human partners than our RSSM baselines.

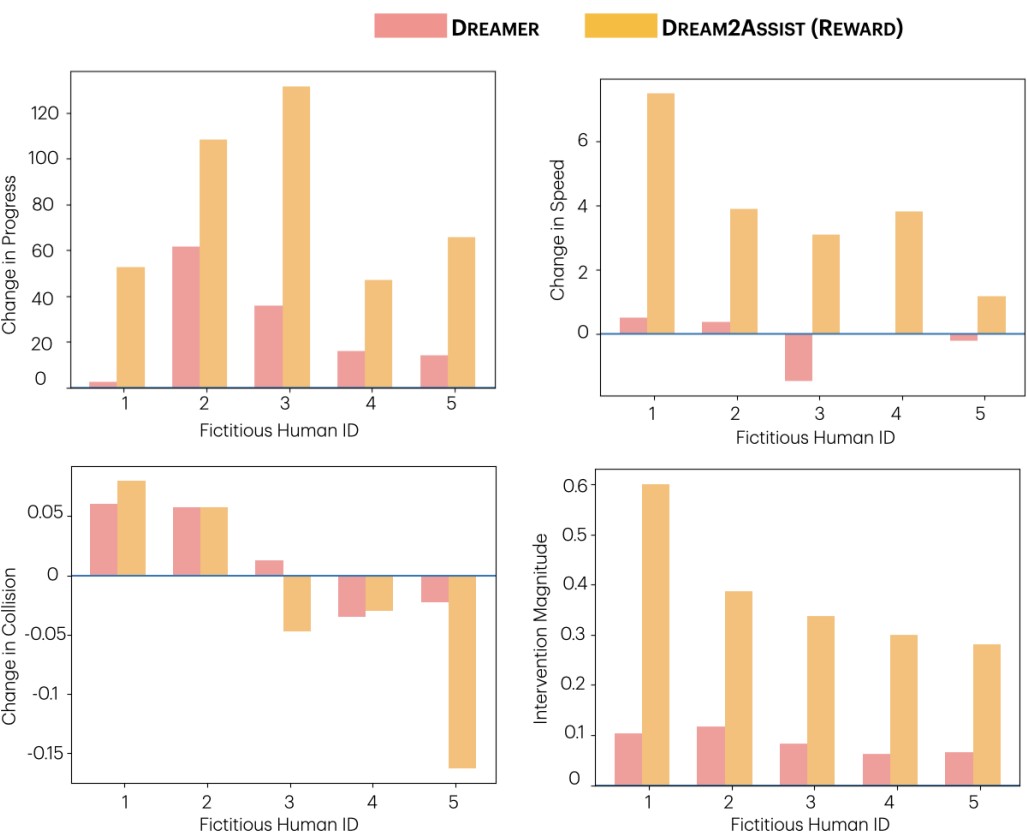

**Figure G.1:** Changes in various metrics in the hairpin scenario when adding assistance to imperfect (1–5) humans whose intent randomly swaps from *stay-behind* to *passing*, averaged over five random seeds and ten trials each. We observe that our inferred-reward objective and our intent-classification objective enable the DREAM2ASSIST agent to generalize much better to this highly out-of-distribution behavior, particularly with significant reductions in collisions and improvements in track progress (i.e., task completion).

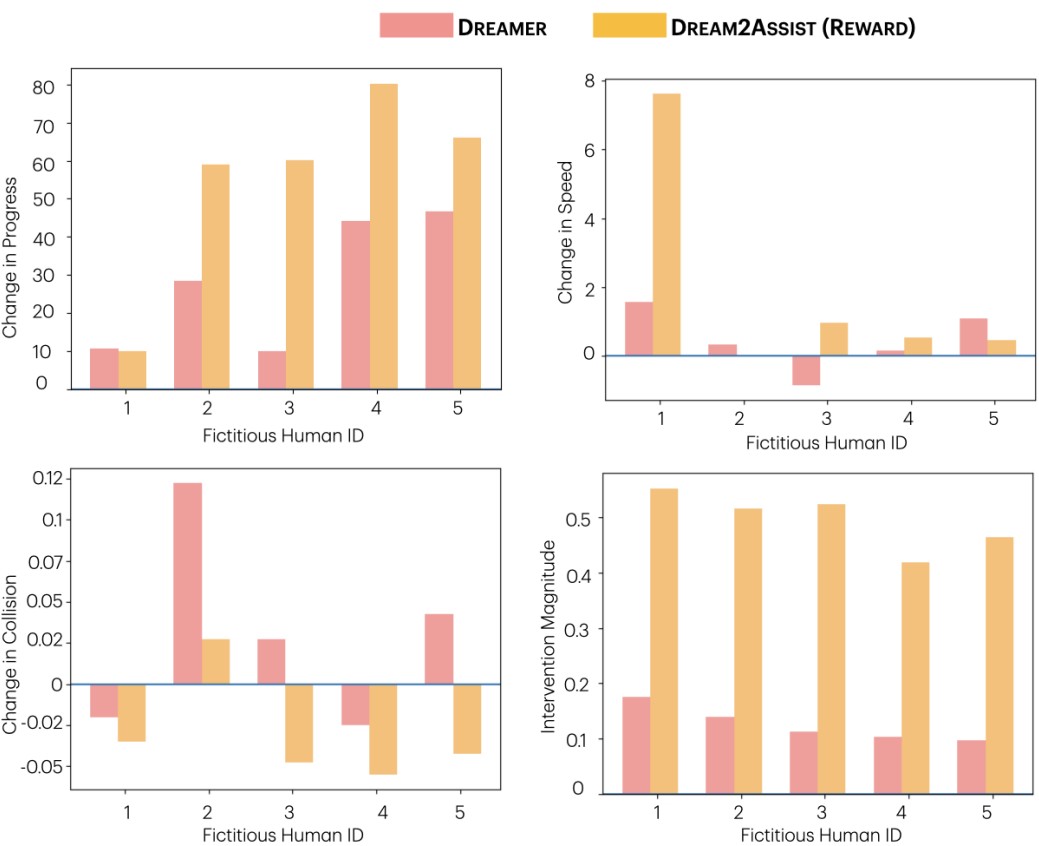

**Figure G.2:** Changes in various metrics in the hairpin scenario when adding assistance to imperfect (1–5) humans whose intent randomly swaps from *pass* to *stay-behind*, averaged over five random seeds and ten trials each. Again, our inferred-reward objective and our intent-classification objective enable the DREAM2ASSIST agent to generalize much better to this highly out-of-distribution behavior, particularly with respect to track progress.

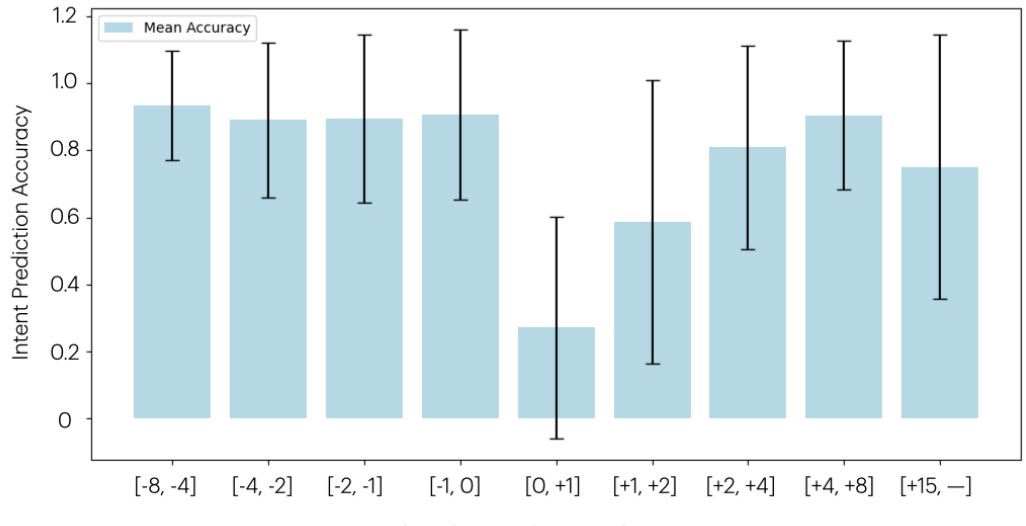

**Figure G.3:** Intent-classification accuracy before and after a synthetic human partner is randomly swapped from a medium-performance *stay-behind* to a medium-performance *pass* partner. We observe that the DREAM2ASSIST agent is able to accurately classify its partner's intent in the 8 seconds leading up to the change, at which point intent-classification accuracy drops sharply. In the following 4 seconds, intent-classification accuracy climbs back up to about 90%, indicating that DREAM2ASSIST is able to recover a human partner's intent even if it is dynamic. The performance dip at the end is likely due to sub-optimalities in the *pass* partner's policy, leading to collisions or spin-outs.

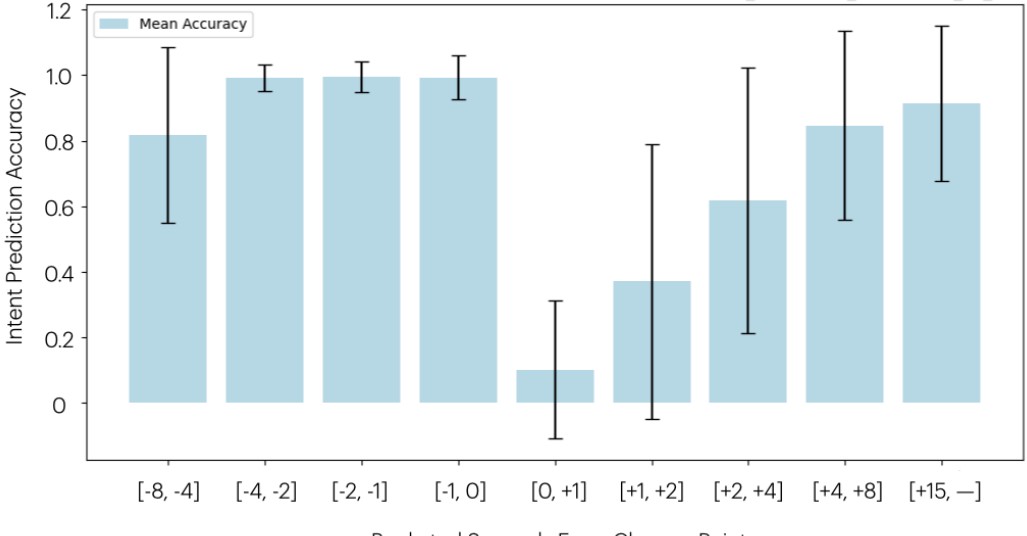

**Figure G.4:** Intent-classification accuracy before and after a synthetic human partner is randomly swapped from a high-performing *pass* to a high-performing *stay-behind* partner. We observe that the DREAM2ASSIST agent is able to accurately classify its partner's intent in the 8 seconds leading up to the change, at which point intent-classification accuracy drops sharply. Intent-classification accuracy climbs monotonically after the change, suggesting that DREAM2ASSIST can perfectly recover intent for high-performing agents, even under unexpected and previously-unseen transitions.