# OpenReview forum: "Dreaming to Assist: Learning to Align with Human Objectives for Shared Control in High-Speed Racing"
_robot-learning.org/CoRL/2024/Conference — CoRL 2024_

### Official Review · Reviewer_TZys · 2024-07-12
**Model-based RL approach for an assistive agent in racing scenarios**

**Originality:** 2
**Technical Quality:** 3
**Clarity Of Presentation:** 2
**Potential Impact:** 2
**Recommendation:** 2
**Confidence:** 3

**Review:**

It was fairly easy to follow the proposed approach. But, the manuscript needs some improvement.

- It was not straightforward to follow Section 4. Some terminologies such as `pass vs no-pass` appear earlier, but the descriptions follow much later. And, Table 1 is presented too early because all other details are described much later as well.
- Are there any plots to show "We observe a consistent trend for all straightaway results – DREAMER offers very low-magnitude intervention, leading to higher performance with expert drivers but poor performance with novice drivers (as the assistant is not helping)."? I believe this is something to show quantitatively. `Figure 3` seems implicitly showing this. Maybe, it's supposed to be in Section 4.1?
- There is inconsistency between `Fig. 4` and `Figure 3`. And, `Fig. 4` appears earlier than `Figure 3`.
- I wasn't conviced to use Dreamer for this work. For example, we can simply train a separate intent predictor to predict user's intention and use the prediction to condition the policy. In this way, the entire architecture could be simplified.  Authors need to explain necessity of the use of model-based RL approach for this problem.

**Quality Of The Limitations Section:**

3

**Questions For Rebuttal:**

- I suggest authors to rewrite Section 4 to re-oreder texts for the reason I described above.
- Authors need to justify the necessity of the use of Dreamer for this work. This feels unnecessary too complicated approach for the given task. It's possible that the simpler approach performs poorly if the racing task is too difficult for such methods, which could convince readers

**Robotics Focus:**

3

**Summary Of Paper:**

This paper proposes a Dreamer-based assistive agent for racing scenarios. The dynamics model is additionally trained to predict intent labels and condition the policy function to assist human drivers towards the aligned objective.

**Summary Of Recommendation:**

The manuscript needs to be improved and authors need to justify the use of Dreamer for this work.

---

### Official Review · Reviewer_5oAj · 2024-07-13
**Minor but solid contribution of a shared control algorithm**

**Originality:** 3
**Technical Quality:** 4
**Clarity Of Presentation:** 3
**Potential Impact:** 2
**Recommendation:** 3
**Confidence:** 5

**Review:**

Strengths:
+ Mostly clear
+ Good, informative figures
+ Novel formulation for learning how to assist in a shared control environment
+ Solid empirical results

Weaknesses:
- Some concepts could be better clarified (e.g., intent, objective, characteristics)
- Limited novelty (mainly, a new reward formulation that includes an ``intent’’)
- Limited evaluation (tested on only one domain, with one set of human distributions)
- Limited discussion over some decisions (representation, hyperparameters)

First, this paper presents a clear contribution: a new formulation of reward with which an assistive agent can learn to perform. It is an incremental work, as it extends an existing approach (DREAM) but clearly explains how it differs from it.
The figures are nicely done and provide good visual aids to better understand the proposed model. The empirical evaluation, while fairly modest, show clear superiority of the proposed approach.

Some concepts could be better explained: intent is the most prominent one. For example, this concept is referred to as a synonym for many different words, but never formally defined. Line 112 states, ``...includes the preferences, desires, and goals of the human (i.e., the human objective),’’ and then ``whose objectives (intents)’’, yet these concepts are not formalized properly anywhere in the texts. Eventually, in terms of the algorithmic representation, “intent” is translated into a portion of the reward that is dedicated to enforcing a certain preference (e.g. to stay in the right lane), which is not a goal.

Similarly, the different “characteristics” that represent the different human agents could be explained and detailed better to help the reader understand how such human agents are expected to differ from one another. Moreover, the conditions ``pass vs. no-pass’’ and ``left vs. right’’ should be explained before they are used.

In terms of related work, there are several papers about shared control and shared tasks with goal monitoring (where intention in these papers is ``goal’’ in the BDI-literature sense):

1. Jiang, Y. S., Warnell, G., & Stone, P. (2021, May). Goal blending for responsive shared autonomy in a navigating vehicle. In Proceedings of the AAAI Conference on Artificial Intelligence (Vol. 35, No. 7, pp. 5939-5947).
2. Macke, W., Mirsky, R., & Stone, P. (2021, May). Expected value of communication for planning in ad hoc teamwork. In Proceedings of the AAAI Conference on Artificial Intelligence (Vol. 35, No. 13, pp. 11290-11298).
3. Dann, M., Yao, Y., Logan, B., & Thangarajah, J. (2021). Multi-Agent Intention Progression with Black-Box Agents. In IJCAI (pp. 132-138).

Also, using ToM to reason about other vehicles has much longer history than the one covered in this work:
4. Koller, D., Weber, J., Huang, T., Malik, J., Ogasawara, G., Rao, B., & Russell, S. (1994, October). Towards robust automatic traffic scene analysis in real-time. In Proceedings of 12th International Conference on Pattern Recognition (Vol. 1, pp. 126-131). IEEE.

Lastly, there is other work on shared control with unknown dynamics:
5. Broad, A., Murphey, T. D., & Argall, B. D. (2017). Learning models for shared control of human-machine systems with unknown dynamics. In 2017 Robotics: Science and Systems, RSS 2017. MIT Press Journals.

In terms of the evaluation, it was unclear what fictitious human drivers were used (were they the same ones used for training?). Also, this portion of the description could be revised: Line 216, ``To measure the contributions of each assistive agent for the fictitious human population, we sample checkpoints at every 20% performance increment for agents up to at least 75% of maximum, or from the bottom five performers if none are under this threshold.’’

Lastly, please go over the citations again to make sure they are all valid. E.g., citation [24] is missing a venue and [62] is missing a year.

*** Post-rebuttal:
I have read the reviews and the author's feedback. In the rebuttal, the authors have answered my questions and have specified changes that address most of my concerns.

**Quality Of The Limitations Section:**

3

**Questions For Rebuttal:**

Q1. Could you explain why the performance of DREAM2ASSIST “left/right” behavior in the straightaway case was not as good as it was in the other cases?
Q2. Figure 3: Are the humans ordered from novice to expert? If not, that might help to see if the claim in Section 4.1 also holds here.
Q3. Have you looked at out-of-distribution behavior of agents? If so, where is it reported? If not, why?
Q4. How were the fictitious human drivers used in training vs. testing? Were they sampled from the same distribution?

**Robotics Focus:**

3

**Summary Of Paper:**

The paper presents a new formulation for an assistive agent in a shared control environment, tested on the CARLA simulator. The new agent is enhanced with an ability to represent the intent of the human teammate, based on several potential pre-learned human types. The paper is clear in most parts, and the empirical evaluation supports the superiority of the proposed approach compared to previous work.

**Summary Of Recommendation:**

There is clear progress in the paper, with both a clear explanation of the algorithm and solid results on two use-cases. However, the contribution is limited.

---

### Official Review · Reviewer_jTCE · 2024-07-21
**Review of submission 417**

**Originality:** 2
**Technical Quality:** 2
**Clarity Of Presentation:** 4
**Potential Impact:** 2
**Recommendation:** 3
**Confidence:** 2

**Review:**

The overall flow of the paper is straightforward to understand, and is well summarized by Figures 1 and 2. The authors ground their work in the setting of shared autonomy for high-speed racing. The method extends the Dreamer RSSM model include a human-intent head that predicts one of several discrete human intents. Given the predicted human intent, the assistive agent is trained to provide a minimal change to the user’s control action to guide the linear combination of the human and agent control to closely match the optimal control for the given intent.

The overall method seems sound, however the authors choose to train and evaluate in simplified settings with easy-to-specify intents that do not change over the interaction. I believe additional experimentation could help understand how the method adapts in closed-loop interaction, as competitive driving scenarios will have interaction effects that cause agents to mutually influence each other.  One potential way to accomplish this would be to simulate opposing agents using a simple ILQ Game solver like in [1] which also investigates the racing setting. In terms of writing, some notations are not defined before use. For example, the symbol x_t is not defined so it isn’t immediately clear whether this means state or observation without prior knowledge of the Dreamer papers. Defining these mathematical symbols could easily be added in Section 2.

The work seems to make an assumption that the human intents (derived from hand-tuned reward functions) used during training sufficiently covers the space of possible human behaviors. The experiments are constructed such that this is easy to do (overtake vs stay behind, right vs left), however I am could imagine the method may struggle in scenarios with a wide range of possible human intents/the policy’s hypothesis space of intents do not cover the full range of potential human behaviors (the hypothesis space misspecification problem [2]).


[1] Hu, Haimin, et al. "Think Deep and Fast: Learning Neural Nonlinear Opinion Dynamics from Inverse Dynamic Games for Split-Second Interactions."
[2] A. Bobu, A. Bajcsy, J. F. Fisac, S. Deglurkar and A. D. Dragan, "Quantifying Hypothesis Space Misspecification in Learning From Human–Robot Demonstrations and Physical Corrections,"


**Update post rebuttal** I thank the authors for taking the time to prepare a clear and concise rebuttal. My clarification questions have been addressed. While I still believe the space of human intents considered is rather narrow, the authors provide sufficient evidence that the assumptions made in this work are fair. The added experiments where human intents change during the interaction makes the method much more compelling.

**Quality Of The Limitations Section:**

3

**Questions For Rebuttal:**

- What happens when the human’s intent changes over time? If the RSSM and intent-predictor head is only trained on humans that maintain fixed intents over time, would this result in the assistive agent struggling to adapt as human intents change? I could see the human’s intent changing in scenarios where the opponent reacts to the human (the opponent isn’t giving me enough space to pass on the left, so I’ll pass on the right instead).
- What was the motivation for using Dreamer as a backbone? The observation space is the privileged state setting with no images. To my knowledge, Dreamer is typically used in settings where you do not have privileged state information and instead only have images (+proprioception). Couldn't you train an intent predictor from a history of past states/actions and ignore the RSSM entirely? Vehicle dynamics are fairly well understood so it is not clear to me why the method needs a latent dynamics model to train the policy either. Is it because high-speed racing induces more complex dynamics (i.e., drifting) that are hard to model?

**Robotics Focus:**

2

**Summary Of Paper:**

This paper proposes a novel method for shared control in high-speed racing scenarios using a modification of the Dreamer world model augmented to predict human intents. The world model and an assistive agent is trained on a suite of simulated humans demonstrating a different human intent (overtaking vs staying behind, right vs left) at varying levels of task proficiency. The authors show in two different simulated racing scenarios that the proposed method outperforms naive Dreamer and imitation learning-based approaches in providing assistance to suboptimal simulated human racers.

**Summary Of Recommendation:**

The paper's evaluation is relatively narrow. The application of high-speed racing is a highly interactive task, yet the evaluation is performed by competing against log-replay data of a human driver. Furthermore, they consider a setting where human intents are fixed when it is quite common for intents (passing on the left vs right) to change over the course of an interaction depending on the behavior of other agents. As it stands, I am not convinced that this method can actually be applied with or around real people.

---

### Author Rebuttal · Authors · 2024-08-13

We thank the reviewers for their effort and insightful comments on our paper. We also thank the reviewers for commenting on the clear writeup of the paper and clear formulation of the problem within an RSSM framework. Indeed, we are looking for a general approach that can address diverse definitions of intent, which is an open research field, as evident from different approaches to the problem in the literature. We will clarify this and highlight relevant example literature in the revised version.

We reiterate some of the most important comments here, in an attempt to further elucidate our presentation, to help clear up any misunderstandings, and to draw connections between reviewer comments and our approaches to address them. We are grateful to engage in a dialogue with the reviewers to clarify further and answer any additional questions.

---

### Decision · Program_Chairs · 2024-09-04

**Decision:**

Accept

**Comment:**

Summary of the Paper

The paper introduces DREAM2ASSIST, a novel framework designed to improve shared control in high-speed racing scenarios by leveraging a modification of the Dreamer world model, which is enhanced to predict human intents. The approach involves training a recurrent state space model (RSSM) that combines both the dynamics of the physical environment and the intentions of human drivers. The assistive agent, based on this model, learns to provide minimal and targeted interventions that align with the predicted human intent, thus optimizing the joint performance of human-robot teams. The effectiveness of this method is demonstrated through simulations in a racing environment using the CARLA simulator, where the model shows improved performance over existing baseline methods.

Strengths

- The paper presents a novel application of the Dreamer world model to shared control scenarios, specifically in high-speed racing. The extension of the model to predict human intents and adapt its policy accordingly is a notable contribution.
- The overall structure of the paper is logical, with well-crafted figures that effectively illustrate the concepts and methodologies proposed. The visual aids, particularly Figures 1 and 2, enhance understanding.
- The proposed method demonstrates superior performance compared to baseline approaches, including naive Dreamer and imitation learning-based models, in simulated environments. The results show that the assistive agent can improve human-robot team performance by aligning interventions with human objectives.
- The work is relevant to the field of robotics, particularly in the context of human-robot interaction and shared autonomy, addressing a challenging domain of high-speed racing.
- The paper clearly addresses the limitations of the current work, providing a foundation for future exploration and improvements.

Weaknesses

- The evaluation of the method is restricted to a specific domain (racing) with a fixed set of human distributions. The lack of generalization to other domains and more complex human intent scenarios is a significant limitation.
- The experimental settings involve simplified scenarios with static human intents, which do not adequately capture the dynamic and interactive nature of real-world racing scenarios where human intents can change over time.
- The necessity of using the Dreamer model in this context is questioned by reviewers. Alternatives such as simpler intent predictors might achieve similar results, and the complexity of the Dreamer-based approach needs better justification.
- Some sections, particularly the experimental setup and results, require reorganization for clarity. Terms such as "intent" and "characteristics" need clearer definitions and earlier introduction to enhance comprehension.
- The reliance on a fixed set of fictitious human drivers during training raises concerns about the model's ability to handle out-of-distribution behaviors and real-world variability in human actions.
- While the approach is innovative, the reviewers perceive the contribution as incremental, extending existing work rather than providing a substantial breakthrough in the field.
- The current work does not include experiments with real human subjects, limiting the applicability and impact of the findings to real-world scenarios.

Summary of the rebuttal phase

While the authors' revisions appear to have improved the quality of the paper, the rebuttal did not change the reviewers' opinions. Although not all reviewers agree, acceptance for a poster presentation is judged to be reasonable.